# Assessing Step-by-Step Reasoning against Lexical Negation: A Case Study on Syllogism

**Mengyu Ye**[1]   **Tatsuki Kuribayashi**[2]   **Jun Suzuki**[1,3]
**Goro Kobayashi**[1,3]   **Hiroaki Funayama**[1,3]
[1]Tohoku University   [2]MBZUAI   [3]RIKEN
{ye.mengyu.s1, goro.koba, h.funa}@dc.tohoku.ac.jp
tatsuki.kuribayashi@mbzuai.ac.ae   jun.suzuki@tohoku.ac.jp

## Abstract

Large language models (LLMs) take advantage of step-by-step reasoning instructions, e.g., chain-of-thought (CoT) prompting. Building on this, their ability to perform CoT-style reasoning robustly is of interest from a probing perspective. In this study, we inspect the step-by-step reasoning ability of LLMs with a focus on negation, which is a core linguistic phenomenon that is difficult to process. In particular, we introduce several controlled settings (e.g., reasoning on fictional entities) to evaluate the logical reasoning abilities of the models. We observed that dozens of modern LLMs were not robust against lexical negation (e.g., *plausible→implausible*) when performing CoT-style reasoning, and the results highlight unique limitations in each LLM family.

https://github.com/muyo8692/stepbystep-reasoning-vs-negation

## 1 Introduction

Few-shot learning (Brown et al., 2020) has led to a remarkable performance in large language models (LLMs). In particular, instructions to generate a reasoning process along with the answer, i.e., chain-of-thought (CoT) prompting (Wei et al., 2022; Kojima et al., 2022), have improved the performance of LLMs. Building on this, the ability of LLMs to perform CoT-style reasoning robustly is of interest from the probing perspective—*how correctly these models perform step-by-step reasoning?*; however, to the best of our knowledge, deeper analyses have yet to be explored fully. To address this question, this study investigates the step-by-step reasoning ability of LLMs with a special focus on robustness against (lexical) negation. Historically, negation has been challenging for neural models (Socher et al., 2013; Kassner and Schütze, 2020), and determining whether the step-by-step reasoning of LLMs overcomes this limitation is important in the natural language processing (NLP) community.

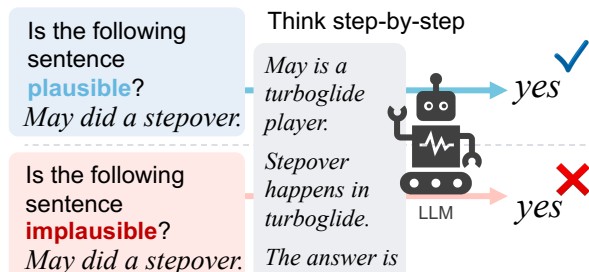

Figure 1: Overview of our experiments conducted to evaluate each model's reasoning ability against lexical negation. The model must answer *no* to the latter question about the **im**plausibility of the sentence based on the valid logical flow. Here, to evaluate the robust logical skill separately, the controlled reasoning chain is given, and the model must derive the answer based solely on the validity of the logical flow without commonsense knowledge due to fictional entities, e.g., *turboglide*.

Our controlled experiments using dozens of LLMs, including GPT-4 (OpenAI, 2023), demonstrate that such models deteriorate in performance substantially when processing questions involving words with just a negative prefix, e.g., *implausible*, *unreasonable* (Figure 1). In addition, the results show that each LLM family has its unique biases against lexical negation, which suggests that different LLM training settings produce substantial differences under certain conditions, and the problems to be addressed are model-dependent. These issues clarify the exact weakness of modern LLMs.

## 2 Reasoning Against Lexical Negation

Given a chain of the reasoning process, we expect the LLM to derive a logically valid conclusion even when the problem involves lexical negation (Figure 1). In Section 2.1, we introduce the task format, and Section 2.2 elaborates on the controlled task settings to elucidate the abilities of the models. Note that our task is similar to CoT reasoning; however, we provide the models with predefined reasoning chains to facilitate controlled analyses.

| Setting | Few-shot exemplars | Target example | If fails at this setting |
|---------|--------------------|-----------------|--------------------------|
| BASE | Is a sentence "A does B" **plausible**? A is a C player. B happens in C/X. So the answer is **yes/no**. | Is a sentence "D does E" **plausible**? D is a F player. E happens in F/Y. So the answer is __ | CoT-style reasoning fails. |
| FIC | Is a sentence "A does B" **plausible**? A is a C player. B happens in C/X. So the answer is **yes/no**. | Is a sentence "$\alpha$ does $\beta$" **plausible**? $\alpha$ is a $\gamma$ player. $\beta$ happens in $\gamma/\chi$. So the answer is __ | Reasoning cannot be abstracted to fictional texts. |
| FICNEG | Is a sentence "A does B" **implausible**? A is a C player. B happens in C/X. So the answer is **yes/no**. | Is a sentence "$\alpha$ does $\beta$" **implausible**? $\alpha$ is a $\gamma$ player. $\beta$ happens in $\gamma/\chi$. So the answer is __ | Abstract CoT-style reasoning is only achieved on the affirmative domain. |
| FICNEG-O | Is a sentence "A does B" **plausible**? A is a C player. B happens in C/X. So the answer is **yes/no**. | Is a sentence "$\alpha$ does $\beta$" **implausible**? $\alpha$ is a $\gamma$ player. $\beta$ happens in $\gamma/\chi$. So the answer is __ | Model cannot handle domain shift in terms of negation. |

Table 1: General task format in each setting. Few-shot exemplars are first shown to a model, and then the model answers to the target example given its question and reasoning chain. Symbols, e.g., A and $\alpha$, are replaced with certain real or fictional entities in the actual input. The REAL setting indicates that the entity choices reflect the factual reality, and FIC. indicates that the entity choices do not reflect factual reality, e.g., *Is "Judy Tate was safe at first." plausible? Judy Tate is a turboglide player. Getting out at first happens in turboglide. So the answer is yes.* Refer to Appendix A for the exact input.

## 2.1 Format: Syllogism

We evaluated the LLMs' ability to judge the validity of particular types of syllogisms. Here, we utilized three settings to ensure the robustness of the results (Section 3); however, we consider the following SPORTS TASK (SP) format as an example to explain the settings. The base format of the syllogism is as follows:

Premise1:  PERSON *is a* SPORT *player.*
Premise2:  ACTION *happens in the* SPORT.
Conclusion: PERSON *does* ACTION.

The above syllogism is converted into instances, as shown in Table 1, comprising a question about the validity of a particular conclusion (*Is a sentence...plausible?*), a chain of the reasoning process (premises), and a *yes/no* answer part. In the experiments, few-shot exemplars (Table 1, column 2) were first input to a model, and then the model completes the answer for the target example (__ in Table 1, column 3) with *yes/no*. Here, the correct answer depends on whether the SPORTS entities mentioned in the chain (premises 1 and 2) are the same.[1] The exact input to the models is described in Appendix A.

## 2.2 Controlled Task Modification

To analyze how the models struggle with negation, we introduce presumably challenging properties into the task gradually (see the examples shown in Table 1).

**BASE setting:** In this setting, premises and conclusions are aligned with the fact, e.g., *Messi did a stepover* is plausible; however, *Messi performed a triple axel* is implausible.

**Fictional setting (FIC):** We do not focus on deriving an answer directly based on the model's knowledge without considering the reasoning chain. To eliminate such a solution from the BASE setting and ablate the effect of factuality, we replace the PERSON and SPORT entities with fictional entities (see Appendix C.1 for details about fictional entities), where the correct conclusion can only be derived from the premise information in a given reasoning chain. Note that these fictional entities are also used in subsequent settings.

**In-domain negation setting (FICNEG):** With this setting, we test the model's robustness against lexical negation. Here, we first design an in-domain setting where both few-shot exemplars and a target example involve lexical negation. Specifically, we turn the original question into one that involves a word with a negative prefix, e.g., *plausible→implausible* (see Appendix B for the word list).[2] Thus, the correct answer to the question

---

[1]Strictly speaking, the answer should also be *unknown* when the two sports differ. In our experiments, our prompts explicate to answer *no* in such cases

[2]Testing negation in syntax, e.g., *not*, is another direction; however, this incurs additional difficulties, e.g., the scope of negation. We adopted the lexical negation as an initial investigation

should be flipped from *yes/no* to *no/yes*.[3]

**Out-domain negation setting (FICNEG-O):**
We design an out-domain setting where few-shot exemplars **do not** involve lexical negation, but the target example has. If a model fails at only this setting, this implies that the model overfits to the domain of the few-shot exemplars in terms of lexical negation. In addition, FIC and FICNEG-O differ only in terms of the existence of the negation in the target example (this point can also be isolated by comparing these results).

## 3 Experimental Settings

**Task:** In addition to the SPORTS TASK (SP) described in Section 2.1, we also introduce several different task formats. One is the OCCUPATION TASK (OC), where the underlying reasoning is similar to the SP task, but the vocabulary and wordings are different:

| | |
|---|---|
| Premise1: | PERSON *is a* TITLE. |
| Premise2: | OCCUPATION *is described as* TITLE. |
| Conclusion: | PERSON *is a* OCCUPATION. |

We also introduce the WEIGHT TRANSITION TASK (WT), where the transitivity between the two propositions is targeted:

| | |
|---|---|
| Premise1: | ANIMAL1 *is heavier than* ANIMAL2. |
| Premise2: | ANIMAL2 *is heavier than* ANIMAL3. |
| Conclusion: | ANIMAL1 *is heavier than* ANIMAL3. |

All of these syllogisms are extended to the four different levels (BASE, FIC, FICNEG, and FICNEG-O) described in Section 2.2. See Appendix C for setting details.

**Data:** For the SP task, we collected 1,000 instances from the sports-understanding task in the BIG-Bench dataset (Srivastava et al., 2022) for the BASE setting,[4] we also manually created the OC and WT instances to be similar to the SP instances.[5] We then modified these instances to create more challenging versions (FIC, FICNEG, and FICNEG-O). To enhance the generality of our findings, we

---

[3]We also adopt a setting involving real entities and negation (NEG) in Appendix D. The results are generally competitive or slightly better than those in the FICNEG setting

[4]Note that the SP task was originally intended to evaluate the commonsense knowledge about sports. In contrast, we used them to assess a pure reasoning ability by providing the necessary facts to derive a conclusion in a reasoning chain.

[5]While we created 1,000 instances for the OC task, 100 instances were created for the WT task since this task is regarded as a supplementary one; nevertheless, quite similar results to the other tasks were obtained.

---

employed 10 variants of the base words and their corresponding negated expressions, e.g., *plausible/implausible*, *reasonable/unreasonable*. Average and standard deviation scores across these runs were reported.

**Models:** We tested 14 common LLMs, e.g., GPT-4 and 3.5 (OpenAI, 2023), four LLaMA variants (Touvron et al., 2023) , Vicuna (Zheng et al., 2023), five OPT varisnts (Zhang et al., 2022), BLOOM (Scao et al., 2022), and BOOMZ (Muennighoff et al., 2022). Additional LLMs are tested in Appendices E and D, including the Alpaca (Taori et al., 2023), OPT-IML (Iyer et al., 2022), GPT-NeoXT (Together Computer, 2023), resulting in a total of 31 LLMs (see Appendix E for more model details).

**Inference:** Three exemplars are given to the model along with general task instructions, e.g., *Let's think step by step* (Appendix A). Note that the exemplars have at least one *yes* and one *no* answer. We also examined different exemplar orders, yielding consistent results independent of the exemplar orders (Appendix F). Here, the answer with the higher probability between *yes* and *no* in the model's outputs for the target example is considered the answer. See Appendix E.1 for additional technical details.

**Metrics:** To evaluate the LLMs, the accuracy of each model's binary answers was measured (see Appendix G for the F1-score results). In addition, to quantify the output bias, we also calculated a *no*-ratio, i.e., how many out of 1,000 instances the models answered *no*. Note that the chance rates of the accuracy and the expected *no*-ratio are 0.5 because the dataset is balanced in terms of the gold answer distribution Appendix C.3).

## 4 Results, Analyses, and Discussions

Tables 2 and 3 show the average and standard deviation of accuracy and *no*-ratio of each model in the SP, OC, and WT tasks.

**Consistent degradation against negation:** We found that all models demonstrated performance degradation with the FICNEG-O setting; however, the GPT-4 model performed above chance (Table 2). In other words, the considered LLMs failed to address lexical negation in CoT-style reasoning. We also observed a notable trend whereby the LLMs preferred to answer *no* regardless of the gold

| Model | Sports Task | | | | Occupation Task | | | | Weight Trans. Task | | | |
|---|---|---|---|---|---|---|---|---|---|---|---|---|
| | Base | Fic | FicNeg | FicNeg-O | Base | Fic | FicNeg | FicNeg-O | Base | Fic | FicNeg | FicNeg-O |
| GPT-4 | 99.0±0.4 | 56.7±4.6 | 92.3±3.3 | 66.6±15.9 | 98.2±0.2 | 76.5±8.1 | 90.2±4.1 | 75.7±12.0 | 100.0±0.0 | 88.6±13.6 | 98.1±5.3 | 77.8±12.1 |
| GPT-3.5 | 99.7±0.1 | 59.8±1.5 | 72.8±4.7 | 36.6±3.5 | 97.1±0.7 | 58.8±1.3 | 58.4±2.5 | 39.9±2.1 | 73.5±4.5 | 68.1±10.0 | 63.6±8.5 | 35.3±10.2 |
| LLaMA-65B | 99.8±0.0 | 89.0±2.9 | 90.7±3.0 | 22.8±16.5 | 100.0±0.0 | 100.0±0.1 | 99.9±0.1 | 15.5±6.3 | 100.0±0.0 | 100.0±0.0 | 66.0±10.1 | 43.6±3.8 |
| LLaMA-30B | 99.8±0.2 | 84.9±3.9 | 99.0±0.5 | 4.9±6.1 | 100.0±0.0 | 99.9±0.1 | 87.9±2.8 | 18.5±13.2 | 99.3±0.8 | 89.3±5.3 | 88.0±8.2 | 44.3±1.5 |
| LLaMA-13B | 98.9±0.4 | 77.1±2.2 | 50.7±1.4 | 23.1±8.4 | 99.9±0.1 | 72.0±5.5 | 91.1±4.8 | 43.6±1.1 | 83.7±5.8 | 91.4±6.3 | 82.2±8.9 | 46.0±0.0 |
| LLaMA-7B | 93.7±1.4 | 63.6±4.8 | 58.6±5.0 | 49.5±0.0 | 68.0±1.8 | 59.7±2.0 | 53.2±2.0 | 46.2±0.0 | 68.2±5.8 | 60.2±4.1 | 57.2±11.6 | 46.0±0.0 |
| Vicuna-13B | 98.4±0.2 | 77.3±1.8 | 83.4±3.7 | 21.6±7.4 | 99.8±0.1 | 72.5±3.3 | 74.0±6.7 | 24.6±5.6 | 70.7±4.0 | 84.8±5.2 | 93.4±2.6 | 40.6±12.4 |
| OPT-175B | 96.5±1.5 | 59.7±5.2 | 62.9±12.8 | 44.5±10.6 | 92.8±1.6 | 92.3±4.7 | 30.2±14.8 | 46.0±0.2 | 80.4±15.2 | 58.0±11.6 | 53.2±7.6 | 41.6±12.9 |
| OPT-66B | 91.7±2.3 | 85.3±4.1 | 35.8±7.2 | 37.4±12.9 | 88.8±2.6 | 99.6±0.4 | 36.9±10.2 | 35.3±11.4 | 86.3±6.6 | 69.9±5.5 | 43.2±2.7 | 46.0±0.0 |
| OPT-30B | 72.5±3.6 | 51.4±0.7 | 47.8±1.8 | 49.2±0.0 | 59.5±1.3 | 54.1±0.3 | 38.6±3.7 | 46.2±0.0 | 54.0±0.0 | 54.0±0.0 | 44.7±3.7 | 46.0±0.0 |
| OPT-13B | 73.3±1.5 | 72.7±6.3 | 49.5±2.7 | 49.2±0.0 | 61.8±2.7 | 58.8±2.7 | 32.4±10.6 | 46.2±0.0 | 77.3±11.0 | 78.3±8.1 | 46.0±0.0 | 46.0±0.0 |
| OPT-6.7B | 85.9±0.8 | 76.5±8.0 | 46.7±3.4 | 45.8±6.6 | 71.4±1.3 | 86.3±2.7 | 26.3±5.1 | 46.2±0.0 | 55.1±1.1 | 54.2±0.4 | 46.0±0.0 | 46.0±0.0 |
| BLOOM | 99.2±0.1 | 89.2±2.7 | 50.5±0.0 | 49.4±0.2 | 100.0±0.1 | 94.1±1.7 | 53.8±0.0 | 46.0±0.3 | 87.6±6.8 | 83.7±8.1 | 50.9±6.8 | 45.4±1.6 |
| BLOOMZ | 91.4±2.0 | 50.5±0.0 | 49.4±0.2 | 48.9±1.3 | 92.0±1.8 | 55.7±0.6 | 45.0±0.6 | 46.2±0.1 | 54.1±0.3 | 54.0±0.0 | 46.0±0.0 | 46.4±1.0 |

Table 2: Average and standard deviation of models' accuracies for each setting in the **Sports Task**, **Occupation Task** and **Weight Trans. Task** (scores are multiplied by 100).

| Model | Sports Task | | | | Occupation Task | | | | Weight Trans. Task | | | |
|---|---|---|---|---|---|---|---|---|---|---|---|---|
| | Base | Fic | FicNeg | FicNeg-O | Base | Fic | FicNeg | FicNeg-O | Base | Fic | FicNeg | FicNeg-O |
| GPT-4 | 50.9±0.5 | 93.8±4.7 | 41.9±3.4 | 77.6±8.0 | 53.2±0.7 | 77.2±8.1 | 36.5±4.2 | 50.1±6.4 | 54.0±0.0 | 65.0±1.4 | 44.1±0.5 | 32.8±1.9 |
| GPT-3.5 | 50.7±0.1 | 90.7±1.5 | 22.3±4.7 | 87.1±3.5 | 55.8±0.8 | 95.0±1.3 | 4.6±2.5 | 93.7±2.1 | 80.3±0.5 | 70.9±1.2 | 73.2±0.9 | 70.9±2.0 |
| LLaMA-65B | 50.2±0.0 | 39.5±2.9 | 58.7±3.1 | 73.2±16.6 | 53.0±0.0 | 53.9±0.1 | 46.2±0.1 | 69.3±6.3 | 54.0±0.0 | 54.0±0.0 | 80.0±1.0 | 97.6±0.4 |
| LLaMA-30B | 50.6±0.2 | 35.4±3.9 | 50.1±0.7 | 50.2±3.3 | 53.0±0.0 | 53.9±0.1 | 34.1±2.8 | 67.4±11.3 | 53.3±0.1 | 43.3±0.5 | 58.0±0.8 | 98.3±0.1 |
| LLaMA-13B | 50.7±0.4 | 48.4±6.2 | 95.2±2.6 | 68.8±12.2 | 53.1±0.1 | 81.8±5.5 | 43.8±9.3 | 97.4±1.1 | 70.3±0.6 | 62.4±0.6 | 62.8±1.0 | 100.0±0.0 |
| LLaMA-7B | 56.6±1.5 | 86.9±4.8 | 10.7±5.5 | 100.0±0.0 | 85.0±1.8 | 94.1±2.0 | 3.1±3.7 | 100.0±0.0 | 85.8±0.6 | 93.8±0.4 | 88.6±1.2 | 100.0±0.0 |
| Vicuna-13B | 50.6±0.4 | 37.0±3.1 | 50.1±9.3 | 37.7±12.6 | 53.2±0.1 | 81.3±3.3 | 20.2±6.7 | 76.5±6.8 | 83.3±0.4 | 69.2±0.5 | 42.0±0.4 | 88.6±0.9 |
| OPT-175B | 46.9±1.5 | 10.2±5.2 | 14.3±12.4 | 95.0±10.8 | 52.1±3.6 | 47.2±5.4 | 41.6±13.2 | 99.8±0.2 | 34.4±1.5 | 12.0±1.2 | 33.2±4.2 | 95.6±1.3 |
| OPT-66B | 47.2±4.8 | 38.4±6.1 | 36.4±22.8 | 88.3±12.8 | 63.9±2.7 | 54.3±0.4 | 17.0±10.4 | 89.1±11.4 | 64.3±0.9 | 83.7±0.6 | 91.8±1.4 | 100.0±0.0 |
| OPT-30B | 77.5±3.6 | 99.4±0.7 | 93.7±4.4 | 100.0±0.0 | 90.3±1.3 | 99.7±0.3 | 90.5±6.8 | 100.0±0.0 | 100.0±0.0 | 100.0±0.0 | 44.3±2.6 | 100.0±0.0 |
| OPT-13B | 55.0±3.0 | 30.4±9.8 | 74.5±13.6 | 100.0±0.0 | 91.2±2.7 | 95.0±2.7 | 75.4±18.4 | 100.0±0.0 | 34.1±1.4 | 45.1±1.9 | 100.0±0.0 | 100.0±0.0 |
| OPT-6.7B | 53.4±2.4 | 31.0±9.4 | 97.3±3.5 | 96.6±6.7 | 77.5±2.7 | 64.6±4.0 | 60.3±17.9 | 100.0±0.0 | 98.9±0.1 | 99.8±0.0 | 100.0±0.0 | 100.0±0.0 |
| BLOOM | 50.1±0.2 | 61.3±2.7 | 0.0±0.0 | 99.9±0.2 | 53.0±0.1 | 59.3±1.9 | 0.0±0.0 | 99.8±0.3 | 64.2±0.9 | 41.7±1.1 | 93.9±1.0 | 98.2±0.5 |
| BLOOMZ | 58.3±2.4 | 100.0±0.0 | 99.9±0.2 | 99.4±1.3 | 60.6±2.0 | 98.1±0.6 | 98.8±0.6 | 99.9±0.2 | 99.9±0.0 | 100.0±0.0 | 100.0±0.0 | 92.4±1.7 |

Table 3: Average and standard deviation of models' *no*-ratio in the model outputs for each setting in the **Sports Task Occupation Task** and **Weight Trans. Task** (scores are multiplied by 100).

answer in the FicNeg-O setting (Table 3). Note that LLMs with accuracy rates of approximately 50% tended to continuously respond with *no* (or *yes*). This finding was particularly noticeable with the FicNeg-O setting where the LLMs that exhibited higher accuracy were those that constantly answered *no* (with the exception of the GPT-4 model). These indicate that models do not randomly behave but exhibit some systematic error patterns. Such consistent degradation was also observed in case of Base→Fic, which suggests that CoT-style prompting is supported by factors aligning with factuality along with the (insufficient) pure reasoning ability of the model.

**Differences across model families:** Interestingly, we also found that different LLM families struggled under different settings (the green to purple patterns in Table 2). For example, the LLaMA models performed well with the FicNeg task but not the OPT models (Table 2). In particular, although the GPT-3.5, OPT-175B, and BLOOM(Z) models have approximately the same scale of parameters, they exhibited contrastive trends. Similar trends were also observed for the *no*-ratio case. For example, with the FicNeg and FicNeg-O, the GPT 3.5, LLaMA-7B, and BLOOM models demonstrated extreme statistics approaching 0 or 100, and their behavior flipped completely due to the different types of prompting between the Fic-

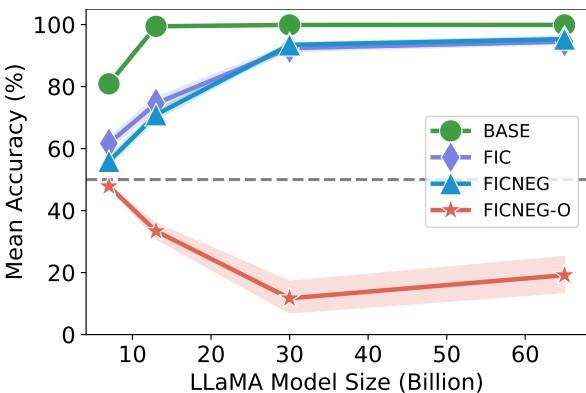

Figure 2: Relationship between model size (x-axis) and performance (y-axis) of LLaMA models for each task. Each point indicates the average accuracy of the corresponding model and setting. The colored area indicates the standard deviation.

NEG and FICNEG-O tasks. The performance difference between, for example, LLaMA-65B and OPT-66B also demonstrates that some factors other than parameter size induce a substantial gap in performance toward certain linguistic phenomena.

**Scaling law breaks:** Scaling law in LLMs has generally been reported (Gordon et al., 2021; Ivgi et al., 2022); however, the improvement over the model scale broke, specifically in the FICNEG-O setting, which confirms that our introduced task is challenging. Figure 2 shows this tendency for the LLaMA models.

In summary, generally, we found that including lexical negation in the tasks caused a drastic performance reduction for the compared LLMs. The results of the controlled experiments further revealed that different LLMs exhibited substantially different limitations and biases. Notably, we further tested the robustness of our findings with different prompt configurations and obtained consistent results (Appendix F).

## 5 Related Work

**Negation and neural models:** Negation is a core operation in natural language and logic, and previous studies have investigated and attempted to improve neural models in terms of addressing negation (Socher et al., 2013; Warstadt et al., 2019; Kim et al., 2019; Kassner and Schütze, 2020; Ettinger, 2020; Hossain et al., 2020; Hosseini et al., 2021; Truong et al., 2023). We align such challenges in the context of CoT-style prompting and the scaling

of LLMs. The closest work to ours reported an inverse scaling law of LLMs' performance against negated prompts (Joel et al., 2023). In addition, we further elucidated the exact limitations and inter-model differences under controlled task settings.

**Step-by-step reasoning:** Generating an inference process with neural models has received increasing attention in terms of both performance improvement and model explainability (Ling et al., 2017; Sun et al., 2019; Rajani et al., 2019; Shwartz et al., 2020; Madaan et al., 2021; Gu et al., 2022; Aoki et al., 2023). Recently, the instruction to make LLMs generate intermediate reasoning steps (i.e., CoT prompting) has led to improvements in model performance (Wei et al., 2022). In this study, we attempted to elucidate the LLM's reasoning ability implicitly assumed in the CoT-style prompting and clarify that this success does not entail the LLMs' robust logical reasoning abilities (particularly against lexical negation). Note that the deterioration in the fictional settings also elcidate that LLMs work well only in the frequent domain in the training data (McCoy et al., 2023).

**Logical reasoning with LLMs and artificially controlled experiments:** Integrating logical reasoning ability into neural models is a pivotal goal in the artificial intelligence field (Marcus, 2003). With this aim, enclosing the models' exact weakness with artificially controlled data has been actively conducted in our field (Betz et al., 2021; Clark et al., 2020; Lu et al., 2021; Kudo et al., 2023); we show the peculiar case that just the flip of one word (adding a nation prefix) causes drastic effects for modern LLMs.

## 6 Conclusions

In this study, we have investigated the ability of LLMs to derive valid conclusions given a reasoning chain with a (lexical) negation, a historically tough phenomenon for neural models. The results of multi-difficulty controlled experiments revealed that LLMs with CoT-style prompting struggled to address negation; a simple flip of one word (e.g., *plausible→implausible*) has significantly hurted their performance. In addition, we have found consistent, systematic failure patterns unique in each LLM family. For example, some models always answered *no* to different question settings. In the future, we plan to analyze the model's internal and explore the source of this weakness.

## Limitations

First, although we considered up to 31 LLMs, several other LLMs cannot be evaluated due to computational limitations, e.g., PaLM-540B (Chowdhery et al., 2022) and PaLM2 (Anil et al., 2023). Thus, evaluating the performance of these models is left to future work. Second, in terms of the generality of the obtained results, the examined prompt variations were limited, although we did examine prompts with different formats and orders (Appendix F). Third, in the current study, we adopted a somewhat peculiar setting where the chain-of-reasoning process is given from the perspective of the original CoT setting. Therefore, exploring the limitations in the inference based on the reasoning chain generated by the model will be an interesting direction from a practical perspective. Fourth, our analysis was limited to behavior-based probing; however, there are other paradigms to investigate (Lasri et al., 2022). In particularly, inspecting the inner workings of the models would be important to understand the mechanism of the model's failure. However, this was difficult because some model parameters were not open, and the vast number of layers/heads/parameters in large models made it difficult to track the precise patterns of the inner workings of the model. Finally, this study only considered lexical negation in English and was further confined to specific task formats and a certain type of syllogism. Therefore, extending the experimental scope will help further elucidate the exact limitations of the models.

## Ethics Statement

Our findings demonstrate that LLMs struggle to address lexical negation under step-by-step CoT-style reasoning settings. This problem is generally related to the problem of hallucinations in LLMs. We hope that our findings help to understand this issue by highlighting their exact weakness against negation.

The synthetic dataset utilized in the current study was created using automatic rules; thus, there were no ethical concerns regarding human workers or annotators during the dataset creation processes. In addition, the entity distribution of the dataset is fully balanced, and most of them are fictional, and there are no intended biases, e.g., the stereotypical relationship between gender and occupation.

## Acknowledgements

We would like to express our gratitude to the members of the Tohoku NLP Group for their insightful comments. And special thanks to Keisuke Sakaguchi for his valuable suggestions on how to improve clarity in several aspects. This work was supported by the JSPS KAKENHI Grant Number JP21H04901, JP22J21492; JST Moonshot R&D Grant Number JPMJMS2011 (fundamental research); and JST SPRING Grant Number JP-MJSP2114.

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

## A  Prompt Examples

Tables 16, 17, 18, and 19 show examples of the exact model input at each setting of our experiments.[6]

## B  Lexical Negation

We created instances involving lexical negation by replacing an adjective in a question with one with a negative prefix. For example, the question *Is the proposition "Messi did a stepover" **plausible**?* is converted to *Is the proposition "Messi did a stepover" **implausible**?* Specifically, we used the terms listed in Table 4 to achieve this conversion. Note that the original SP task only adopts the word *plausible*. Here, we enhanced the diversity of the prompts to ensure the generality of our findings. The lexical negation list was created as follows: (i) GPT-4 was employed to generate nine synonyms of the word *plausible*, and then (ii) we manually added proper negation prefixes to each synonym to form the lexically negated term.

## C  Task Details

Here, we describe the task settings in detail. To ensure the robustness of our findings, we conduct additional experiments on two tasks (in addition to the SP task), i.e., the OCCUPATION (OC) and weight transitivity (WEIGHTTRANS.; WT) tasks. The results across the tasks support the overall conclusion derived in the main part of this paper (See Appendix D for additional information).

### C.1  Fictional Names/Information

In the fictional settings, we used fictional entities in all tasks. Here, we used GPT-4 to generate the names of the fictional sports, occupations, and animals. We used five fictional sports (i.e., (*hydrosprint*, *aeropaddleball*, *gravitydodge*, *turboglide* and *titantumble*)), five fictional occupations (i.e., *hydropurator*, *sonotextilist*, *chronoarchaeor*, *quantumbotanialist*, and *psycostylist*) and

| Base words | Negated words |
|---|---|
| *plausible* | *implausible* |
| *believable* | *unbelievable* |
| *reasonable* | *unreasonable* |
| *thinkable* | *unthinkable* |
| *probable* | *improbable* |
| *imaginable* | *unimaginable* |
| *convincing* | *unconvincing* |
| *conceivable* | *inconceivable* |
| *feasible* | *unfeasible* |
| *credible* | *uncredible* |

Table 4: Full list of base words and lexically negated words used in the experiments.

1,217 fictional animals (see Table 5 for specific examples). In terms of people's names, we initially collected 50 typical male and female first names from the "Name Corpus: List of Male, Female, and Pet names" [7], which is available in the CMU Artificial Intelligence Repository. We then randomly collected 100 family names from the "TelligentCommunitySample 0.1.1" dataset, [8] which is accessible via the PowerShell Gallery. Finally, we created a list of 100 fictional names for each sport by randomly combining the first and last names (examples for the SPORTS task are shown in Table 6). We also used the weight data of Mammals [9] to generate the gold label in the BASE (non-fictional) setting of the WEIGHT TRANS. task.

### C.2  Task Formats

**SPORTS Task:**  See Section 2 and Table 1.

**OCCUPATION Task:**  The task format in the FICNEG-O setting is described as follows:

> Few-shot exemplar:
> Q: Is a sentence "PERSON is a TITLE" **plausible**?
> A: PERSON is a OCCUPATION1. Only OCCUPATION1/2 are TITLE. So the answer is **yes/no**.
>
> Target example:
> Q: Is a sentence "PERSON is a TITLE" **implausi-**

---

[6]In consideration of readability, this is presented on the last page of the paper.

[7]https://www.cs.cmu.edu/Groups/AI/util/areas/nlp/corpora/names/0.html

[8]https://www.powershellgallery.com/packages/TelligentCommunitySample/0.1.1

[9]Mammals ordered by their weight: https://thewebsiteofeverything.com/animals/mammals/adult-weight.html

| Fictional Sports | Fictional Occupations | Fictional Animals | | |
|---|---|---|---|---|
| Hydrosprint | Hydropurator | Flickerbeast | Striped Quillaphant | Whiskerfluff |
| Aeropaddleball | Sonotextilist | Quokkalinga | Pinnapartholanka | Glidefin Skyweasel |
| Gravitydodge | Chronoarchaeor | Prismazebra | Pangolirex | Fawnimouse |
| Turboglide | Quantumbotanialist | WaveSkitterer | Shadow Glidehopper | Nimbuswolftail |
| Titantumble | Psycostylist | Fluffentinger | Glimmerhorn Crestail | Grizmalian Whiskerlop |

Table 5: Examples of fictional sports, occupations and animals we used.

| Hydrosprint | Aeropaddleball | Gravitydodge | Turboglide | Titantumble |
|---|---|---|---|---|
| *Tilda Pruitt* | *Phoebe Richardson* | *Hussein Whitfield* | *Rob Hancock* | *Vita Elmore* |
| *Sansone Brady* | *Michel Allen* | *Larisa Keller* | *Chas Morrow* | *Jay Fowler* |
| *Judy Tate* | *Alyssa McIntyre* | *Wilburn Anderson* | *Sonja Fletcher* | *Stefan Camp* |
| *Petrina Norman* | *Dosi Sykes* | *Ernesto Hall* | textit*Kaleb Graham* | *Malcolm Pearson* |
| *Rutherford Lucas* | *Francisco McCoy* | *Douggie Barbour* | *Garwin Shields* | *Cassi Cooke* |
| *Way Franklin* | *Lorain Reid* | *Celia Jain* | *Gunter Payne* | *Linnet Page* |
| *Jannel Stanton* | *Neda Rose* | *Raynard Kemp* | *Elliott Blum* | *Myrilla Anderson* |
| *Ora Law* | *Sonni Burnett* | *Gregor O'Neill* | *Hailey Hatcher* | *Tobye Washington* |
| *Owen McGee* | *Agathe Frederick* | *Carlton Morris* | *Cornelius McCarthy* | *Granville White* |
| *Kalvin Barr* | *Darrick Rogers* | *Katti Davies* | *Parker Baxter* | *Corny Reid* |

Table 6: Examples of fiction person names we used for each fictional sport.

ble?
A: PERSON is a OCCUPATION1. Only OCCUPATION1/2 are TITLE. So the answer is __

Put simply, the underlying reasoning flow is similar to that of the SP task; however, here, the entities (i.e., the occupation and property names) differ.

**WEIGHT TRANS. Task:** The task format in the FICNEG-O setting is as follows:

Few-shot exemplar:
Is a sentence "ANIMAL1 is heavier than ANIMAL2" **plausible**?
ANIMAL1/2 is heavier than ANIMAL3. ANIMAL3 is heavier than ANIMAL2/1. So the answer is **yes/no**.

Target example:
Is a sentence "ANIMAL1 is heavier than ANIMAL2" **implausible**?
ANIMAL1/2 is heavier than ANIMAL3. ANIMAL3 is heavier than ANIMAL2/1. So the answer is __

Here, the transitivity of reasoning (A>B, B>C, then A>C) is targeted.

### C.3 Answer Distribution

Essentially, the *yes*:*no* ratio of the gold labels was approximately 1:1. Strictly speaking, the distribution differed slightly from 1:1 due to the random seed used in the dataset creation process. For example, for the SPORTS task, the BASE dataset in-

cluded 496 *yes* labels and 504 *no* labels, and the FIC dataset included 495 *yes* labels and 505 *no* labels. The FICNEG dataset included 504 *yes* labels and 496 *no* labels, and the FICNEG-O dataset included 505 *yes* labels and 495 *no* labels.

## D Full Results

All results for the SP, OC, and WT tasks are shown in Table 9, 10, and 11, respectively[10]. Note that the WT experiment was conducted at a 1/10 scale (1,000 instances=100 seed instances×10 negated words) as a supplementary experiment.

We also examined the textscNeg setting, where real (not fictional) entities were used; however, the question involved negation as an intermediate setting between the BASE and FICNEG settings. The performance of all models is shown in Table 8. As can be seen, the results are generally competitive or slightly better than those obtained with the FICNEG setting. In other words, the model cannot handle negation in natural text, and abstract reasoning over negation is even more difficult.

## E Models

In this study, we evaluated the 31 models listed in Table 15.[11] For the GPT-4 (gpt-4-0314) (OpenAI, 2023), and GPT-3.5 models (i.e., text-davinci-002,

---

[10]In consideration of readability, these tables are presented after several tables

[11]Presented after several tables demonstrating supplemental results.

text-davinci-003 (Ouyang et al., 2022), and gpt-3.5-turbo-0301), the experiments were conducted on June 2023 utilizing OpenAI's API. Note that gpt-4-0314 and gpt-3.5-turbo-0301 will be phased out in the near future.

The experiments for the other (non-OpenAI) models were conducted using Huggingface Transformers (Wolf et al., 2020) with the 8-bit option (Dettmers et al., 2022). For the LLaMA (Touvron et al., 2023) models, we received the model weights from the LLAMA Release Team on May 25, 2023. In addition, we recovered the Vicuna and Alpaca (Taori et al., 2023) models based on the provided LLaMA weights. For the OPT models ranging from 1.3 B to 66B, OPT-IML models, and OPT-IML-Max models (Iyer et al., 2022), we employed the models available from the Huggingface Community Model Hub[12]. We received the model weight for the OPT-175B (Zhang et al., 2022) model from Meta on May 28, 2022. We also used the BLOOM (Scao et al., 2022), BLOOMZ (Muennighoff et al., 2022), and NeoXT-Chat-Base-20B (Together Computer, 2023) models available from the Huggingface Community Model Hub.

### E.1 Model Settings During Generation

To ensure that the models only output *yes* or *no*, we applied some changes during the answer generation process. Specifically, for the OpenAI models, we introduced an equal logit bias to *yes* and *no* using the provided $logit\_bias$ option, while setting $tempreture = 0.0, max\_tokens = 1$. For the other non-OpenAI models, we manually ascertained the logit of *yes* and *no*, ultimately using the greater of the two as the model's final response under the same settings as the OpenAI models, in which $temperature = 0.0, max\_new\_tokens = 1$.

### F Robustness over Different Prompts

To ensure the robustness of our results across different settings, we conducted supplementary experiments to investigate both prompt order and format. These experiments were conducted using the SP and OC tasks. Note that these supplementary experiments were conducted at 1/10 scale (1,000 instances=100 seed instances×10 negated words).

**Fictional prompt:** The few-shot exemplars in the main experiments consistently involved real

[12] https://huggingface.co/models

| Model | Accuracy | No-ratio |
|---|---|---|
| GPT-4 | 99.8 ±0.3 | 52.1 ±1.8 |
| GPT-3.5-turbo | 90.5 ±2.4 | 61.6 ±1.7 |
| text-davinci-003 | 99.5 ±0.4 | 52.0 ±1.6 |
| text-davinci-002 | 99.3 ±0.3 | 52.9 ±1.8 |
| LLaMA-65B | 100.0 ±0.0 | 52.1 ±1.7 |
| LLaMA-30B | 99.5 ±0.5 | 51.6 ±1.5 |
| LLaMA-13B | 95.3 ±3.3 | 50.0 ±6.1 |
| LLaMA-7B | 85.1 ±3.1 | 62.3 ±8.3 |
| Vicuna-13B | 93.7 ±5.3 | 47.2 ±8.3 |
| Vicuna-7B | 93.4 ±3.7 | 48.5 ±7.8 |
| Alpaca-7B | 89.5 ±3.2 | 52.5 ±3.6 |
| OPT-175B | 65.6 ±14.4 | 17.7 ±12.7 |
| OPT-66B | 93.0 ±3.5 | 52.0 ±3.0 |
| OPT-30B | 58.0 ±3.2 | 87.8 ±10.6 |
| OPT-13B | 64.1 ±4.9 | 71.8 ±7.6 |
| OPT-6.7B | 71.8 ±4.1 | 47.1 ±5.5 |
| OPT-2.7B | 55.7 ±2.5 | 83.3 ±18.6 |
| OPT-1.3B | 67.2 ±5.5 | 55.6 ±21.3 |
| OPT-IML-Max-30B | 84.7 ±2.3 | 46.6 ±12.3 |
| OPT-IML-Max-1.3B | 62.1 ±6.2 | 76.6 ±12.3 |
| OPT-IML-30B | 80.4 ±4.4 | 39.0 ±11.9 |
| OPT-IML-1.3B | 65.3 ±5.4 | 67.4 ±12.2 |
| BLOOM | 94.5 ±2.1 | 46.7 ±3.7 |
| BLOOM-7.1B | 58.7 ±1.9 | 35.0 ±8.3 |
| BLOOM-3B | 56.4 ±0.7 | 93.8 ±3.9 |
| BLOOM-1.7B | 50.2 ±3.1 | 2.9 ±2.6 |
| BLOOMZ | 76.0 ±4.4 | 70.5 ±9.9 |
| BLOOMZ-7.1B | 54.0 ±2.8 | 98.1 ±1.4 |
| BLOOMZ-3B | 52.7 ±2.3 | 99.4 ±0.6 |
| BLOOMZ-1.7B | 52.3 ±1.8 | 99.8 ±0.3 |
| NeoXT-Chat-Base-20B | 73.4 ±9.7 | 78.5 ±9.3 |

Table 7: Average and standard deviation of models' accuracies and the *no*-ratio of the model outputs in **Fictional Prompt**.

entities. Thus, we conducted supplementary experiments in which the few-shot exemplars pertained to fictional entities. These experiments were implemented under the FIC setting, and the results are presented in Table 7, where the values are the averages from the SP and OC tasks.

**Prompt format:** We explored the influence of the prompt format in both few-shot exemplars and target examples. Here, we used the following format on questions with the gold labels designated as *no*. (Note, the format with gold labels of *yes* were unaltered.) A corresponding example is shown as follows:

Is a sentence "PERSON does ACTION" **plausible**?
PERSON is a SPORTS palyer.
ACTION happens/does not happen in SPORTS.
So the answer is **yes/no**.

| Model | Accuracy | No-ratio |
|---|---|---|
| GPT-4 | 91.1 ±8.8 | 56.1 ±7.8 |
| GPT-3.5-turbo | 86.1 ±13.4 | 40.2 ±9.1 |
| text-davinci-003 | 99.6 ±0.5 | 48.6 ±1.1 |
| text-davinci-002 | 100.0 ±0.0 | 48.3 ±1.3 |
| LLaMA-65B | 99.8 ±0.2 | 48.2 ±1.3 |
| LLaMA-30B | 96.6 ±2.0 | 44.9 ±3.0 |
| LLaMA-13B | 83.3 ±9.7 | 62.2 ±10.8 |
| LLaMA-7B | 70.0 ±8.5 | 30.0 ±9.2 |
| Vicuna-13B | 97.4 ±1.7 | 49.1 ±1.7 |
| Vicuna-7B | 82.3 ±14.6 | 31.1 ±16.4 |
| Alpaca-7B | 55.4 ±15.3 | 71.0 ±9.4 |
| OPT-175B | 61.7 ±11.2 | 31.1 ±10.7 |
| OPT-66B | 31.8 ±5.7 | 38.9 ±14.4 |
| OPT-30B | 42.3 ±4.4 | 93.2 ±5.0 |
| OPT-13B | 47.2 ±3.7 | 63.5 ±13.0 |
| OPT-6.7B | 45.2 ±5.8 | 71.4 ±20.0 |
| OPT-2.7B | 45.8 ±4.4 | 35.3 ±20.6 |
| OPT-1.3B | 48.4 ±1.3 | 99.2 ±1.2 |
| OPT-IML-Max-30B | 42.3 ±10.5 | 14.2 ±12.6 |
| OPT-IML-Max-1.3B | 48.3 ±1.4 | 100.0 ±0.2 |
| OPT-IML-30B | 20.8 ±11.8 | 44.7 ±9.2 |
| OPT-IML-1.3B | 48.2 ±1.5 | 99.7 ±0.4 |
| BLOOM | 63.7 ±6.0 | 16.1 ±7.7 |
| BLOOM-7.1B | 55.1 ±2.2 | 11.7 ±8.0 |
| BLOOM-3B | 52.0 ±1.7 | 0.5 ±0.4 |
| BLOOM-1.7B | 48.4 ±1.3 | 99.9 ±0.1 |
| BLOOMZ | 17.5 ±6.5 | 65.3 ±10.0 |
| BLOOMZ-7.1B | 48.1 ±1.1 | 99.8 ±0.2 |
| BLOOMZ-3B | 48.0 ±1.5 | 99.5 ±0.4 |
| BLOOMZ-1.7B | 48.3 ±1.3 | 100.0 ±0.0 |
| NeoXT-Chat-Base-20B | 49.3 ±1.9 | 96.0 ±3.4 |

Table 8: Average and standard deviation of model accuracies and *no*-ratio for the NEG setting (i.e., real entities and questions with negation).

---

Is a sentence "PERSON does ACTION" **implausible**?
PERSON is a SPORTS palyer.
ACTION happens/does not happen in SPORTS.
So the answer is **no/yes**.

---

Compared to the original format (Section 2), premise 2 changes. Here, the task is not to identify the consistency of sports/occupation name; however, the conclusion depends on the existence of *does not*.

The results are shown in Table 12. Note that both the accuracy and no ratio values are the averages obtained from the SP and OC tasks.

**Prompt order:** We investigated the impact of the prompt order with a specific focus on the position of the *no* label in the three exemplars. The order of the three exemplars in the main experiments

was *yes, no, yes*; thus, we conducted supplemental experiments where the gold label sequences were altered to *yes, yes, no* and *no, yes, yes*. The results of the prompt order experiments are shown in Table 13, which shows the averages from the SP and OC tasks.

## G  F1 Score

Certain models (e.g., BLOOMZ family and OPT family in Table 9) predominantly registered an accuracy of approximately 50% by consistently responding with *no* (or *yes*). Note that this pattern was particularly evident for the FICNEG-O setting, with the GPT-4 model being a significant outlier. To highlight these models, we provided the macro-averaged F1-scores in Table 14.

| Model | Accuracy | | | | No-ratio | | | |
|---|---|---|---|---|---|---|---|---|
| | BASE | FIC | FICNEG | FICNEG-O | BASE | FIC | FICNEG | FICNEG-O |
| GPT-4 | 99.0±0.4 | 56.7±4.6 | 92.3±3.3 | 66.6±15.9 | 50.9±0.5 | 93.8±4.7 | 41.9±3.4 | 77.6±8.0 |
| GPT-3.5-turbo | 99.7±0.1 | 59.8±1.5 | 72.8±4.7 | 36.6±3.5 | 50.7±0.1 | 90.7±1.5 | 22.3±4.7 | 87.1±3.5 |
| text-davinci-003 | 99.9±0.1 | 81.6±1.8 | 87.2±3.2 | 28.0±9.6 | 50.4±0.1 | 40.4±3.3 | 51.2±6.7 | 64.8±8.2 |
| text-davinci-002 | 100.0±0.0 | 74.4±2.8 | 91.1±2.3 | 49.1±0.3 | 50.4±0.0 | 76.0±2.7 | 40.6±2.2 | 99.8±0.3 |
| LLaMA-65B | 99.8±0.0 | 89.0±2.9 | 90.7±3.0 | 22.8±16.5 | 50.2±0.0 | 39.5±2.9 | 58.7±3.1 | 73.2±16.6 |
| LLaMA-30B | 99.8±0.2 | 84.9±3.9 | 99.0±0.5 | 4.9±6.1 | 50.6±0.2 | 35.4±3.9 | 50.1±0.7 | 50.2±3.3 |
| LLaMA-13B | 98.9±0.4 | 77.1±2.2 | 50.7±1.4 | 23.1±8.4 | 50.7±0.4 | 48.4±6.2 | 95.2±2.6 | 68.8±12.2 |
| LLaMA-7B | 93.7±1.4 | 63.6±4.8 | 58.6±5.0 | 49.5±0.0 | 56.6±1.5 | 86.9±4.8 | 10.7±5.5 | 100.0±0.0 |
| Vicuna-13B | 98.4±0.2 | 77.3±1.8 | 83.4±3.7 | 21.6±7.4 | 50.6±0.4 | 37.0±3.1 | 50.1±9.3 | 37.7±12.6 |
| Vicuna-7B | 98.3±0.3 | 93.0±2.6 | 58.5±6.8 | 29.7±20.2 | 50.3±0.8 | 57.4±2.6 | 8.1±6.9 | 52.0±25.9 |
| Alpaca-7B | 91.4±1.7 | 83.3±4.9 | 48.6±3.4 | 49.5±1.7 | 43.8±2.6 | 67.0±5.0 | 85.6±8.2 | 98.2±3.0 |
| OPT-175B | 96.5±1.5 | 59.7±5.2 | 62.9±12.8 | 44.5±10.6 | 46.9±1.5 | 10.2±5.2 | 14.3±12.4 | 95.0±10.8 |
| OPT-66B | 91.7±2.3 | 85.3±4.1 | 35.8±7.2 | 37.4±12.9 | 47.2±4.8 | 38.4±6.1 | 36.4±22.8 | 88.3±12.8 |
| OPT-30B | 72.5±3.6 | 51.4±0.7 | 47.8±1.8 | 49.2±0.0 | 77.5±3.6 | 99.4±0.7 | 93.7±4.4 | 100.0±0.0 |
| OPT-13B | 73.3±1.5 | 72.7±6.3 | 49.5±2.7 | 49.2±0.0 | 55.0±3.0 | 30.4±9.8 | 74.5±13.6 | 100.0±0.0 |
| OPT-6.7B | 85.9±0.8 | 76.5±8.0 | 46.7±3.4 | 45.8±6.6 | 53.4±2.4 | 31.0±9.4 | 97.3±3.5 | 96.6±6.7 |
| OPT-2.7B | 75.2±5.4 | 54.4±5.4 | 35.5±6.0 | 39.9±9.5 | 42.2±11.8 | 7.9±7.4 | 36.2±22.0 | 80.5±24.4 |
| OPT-1.3B | 70.7±4.8 | 56.6±4.6 | 49.2±0.0 | 47.7±4.7 | 72.0±8.8 | 20.1±15.2 | 100.0±0.0 | 96.4±11.4 |
| OPT-IML-Max-30B | 96.1±0.4 | 80.8±4.5 | 51.8±4.6 | 46.5±5.5 | 51.4±1.6 | 69.5±4.8 | 10.3±14.7 | 73.2±12.9 |
| OPT-IML-Max-1.3B | 57.7±1.5 | 51.7±1.4 | 49.5±0.0 | 49.5±0.0 | 92.3±1.9 | 98.8±1.4 | 100.0±0.0 | 100.0±0.0 |
| OPT-IML-30B | 94.8±0.7 | 83.3±5.8 | 52.2±8.9 | 46.3±2.8 | 49.1±1.9 | 66.1±7.0 | 50.2±15.2 | 91.7±3.7 |
| OPT-IML-1.3B | 58.1±0.8 | 52.9±1.4 | 49.5±0.0 | 49.5±0.0 | 89.4±2.0 | 97.6±1.4 | 100.0±0.0 | 100.0±0.0 |
| BLOOM | 99.2±0.1 | 89.2±2.7 | 50.5±0.0 | 49.4±0.2 | 50.1±0.2 | 61.3±2.7 | 0.0±0.0 | 99.9±0.2 |
| BLOOM-7.1B | 68.3±0.9 | 50.9±1.0 | 50.5±0.0 | 43.6±9.8 | 55.4±6.9 | 2.4±1.4 | 0.0±0.0 | 73.4±39.4 |
| BLOOM-3B | 51.4±0.6 | 50.5±0.0 | 51.5±0.7 | 49.5±0.0 | 99.0±0.6 | 100.0±0.0 | 2.4±2.1 | 100.0±0.0 |
| BLOOM-1.7B | 50.7±0.9 | 49.5±0.0 | 49.5±0.0 | 49.5±0.0 | 2.3±2.4 | 0.0±0.0 | 100.0±0.0 | 100.0±0.0 |
| BLOOMZ | 91.4±2.0 | 50.5±0.0 | 49.4±0.2 | 48.9±1.3 | 58.3±2.4 | 100.0±0.0 | 99.9±0.2 | 99.4±1.3 |
| BLOOMZ-7.1B | 52.8±1.0 | 50.5±0.0 | 49.5±0.0 | 49.5±0.0 | 97.5±1.1 | 100.0±0.0 | 100.0±0.0 | 100.0±0.0 |
| BLOOMZ-3B | 50.7±0.2 | 50.5±0.0 | 49.5±0.0 | 49.5±0.0 | 99.6±0.2 | 100.0±0.0 | 100.0±0.0 | 100.0±0.0 |
| BLOOMZ-1.7B | 50.6±0.3 | 50.5±0.0 | 49.5±0.0 | 49.5±0.0 | 99.8±0.3 | 100.0±0.0 | 100.0±0.0 | 100.0±0.0 |
| NeoXT-Chat-Base-20B | 77.5±3.8 | 52.4±1.8 | 49.5±0.1 | 49.5±0.0 | 72.9±3.8 | 98.1±1.8 | 100.0±0.1 | 100.0±0.0 |

Table 9: Average and standard deviation of model accuracies and the no-ratio of the model outputs at each setting for the SPORTS TASK.

| Model | Accuracy | | | | No-ratio | | | |
|---|---|---|---|---|---|---|---|---|
| | BASE | FIC | FICNEG | FICNEG-O | BASE | FIC | FICNEG | FICNEG-O |
| GPT-4 | 98.2±0.2 | 76.5±8.1 | 90.2±4.1 | 75.7±12.0 | 53.2±0.7 | 77.2±8.1 | 36.5±4.2 | 50.1±6.4 |
| GPT-3.5-turbo | 97.1±0.7 | 58.8±1.3 | 58.4±2.5 | 39.9±2.1 | 55.8±0.8 | 95.0±1.3 | 4.6±2.5 | 93.7±2.1 |
| text-davinci-003 | 99.9±0.0 | 73.3±2.4 | 60.6±2.6 | 25.9±4.3 | 53.1±0.0 | 80.5±2.4 | 6.9±2.7 | 77.9±1.7 |
| text-davinci-002 | 100.0±0.0 | 63.7±3.0 | 62.9±4.2 | 49.3±6.3 | 53.0±0.0 | 90.1±3.0 | 9.1±4.2 | 95.7±6.0 |
| LLaMA-65B | 100.0±0.0 | 100.0±0.1 | 99.9±0.1 | 15.5±6.3 | 53.0±0.0 | 53.9±0.1 | 46.2±0.1 | 69.3±6.3 |
| LLaMA-30B | 100.0±0.0 | 99.9±0.1 | 87.9±2.8 | 18.5±13.2 | 53.0±0.0 | 53.9±0.1 | 34.1±2.8 | 67.4±11.3 |
| LLaMA-13B | 99.9±0.1 | 72.0±5.5 | 91.1±4.8 | 43.6±1.1 | 53.1±0.1 | 81.8±5.5 | 43.8±9.3 | 97.4±1.1 |
| LLaMA-7B | 68.0±1.8 | 59.7±2.0 | 53.2±2.0 | 46.2±0.0 | 85.0±1.8 | 94.1±2.0 | 3.1±3.7 | 100.0±0.0 |
| Vicuna-13B | 99.8±0.1 | 72.5±3.3 | 74.0±6.7 | 24.6±5.6 | 53.2±0.1 | 81.3±3.3 | 20.2±6.7 | 76.5±6.8 |
| Vicuna-7B | 93.5±2.5 | 64.4±2.5 | 53.8±0.0 | 38.3±5.2 | 59.5±2.5 | 89.4±2.5 | 0.0±0.0 | 82.2±26.1 |
| Alpaca-7B | 83.4±2.9 | 83.4±4.5 | 39.2±6.0 | 43.5±1.7 | 69.6±2.9 | 70.0±4.7 | 78.0±9.7 | 97.2±1.7 |
| OPT-175B | 92.8±1.6 | 92.3±4.7 | 30.2±14.8 | 46.0±0.2 | 52.1±3.6 | 47.2±5.4 | 41.6±13.2 | 99.8±0.2 |
| OPT-66B | 88.8±2.6 | 99.6±0.4 | 36.9±10.2 | 35.3±11.4 | 63.9±2.7 | 54.3±0.4 | 17.0±10.4 | 89.1±11.4 |
| OPT-30B | 59.5±1.3 | 54.1±0.3 | 38.6±3.7 | 46.2±0.0 | 93.0±1.3 | 99.7±0.3 | 90.5±6.8 | 100.0±0.0 |
| OPT-13B | 61.8±2.7 | 58.8±2.7 | 32.4±10.6 | 46.2±0.0 | 91.2±2.7 | 95.0±2.7 | 75.4±18.4 | 100.0±0.0 |
| OPT-6.7B | 71.4±1.3 | 86.3±2.7 | 26.3±5.1 | 46.2±0.0 | 77.5±2.7 | 64.6±4.0 | 60.3±17.9 | 100.0±0.0 |
| OPT-2.7B | 53.4±0.3 | 55.0±0.9 | 45.1±7.3 | 46.2±0.0 | 99.6±0.4 | 98.8±1.0 | 10.7±9.4 | 100.0±0.0 |
| OPT-1.3B | 57.2±1.2 | 66.9±6.2 | 46.1±0.1 | 46.2±0.0 | 95.6±1.5 | 86.8±6.3 | 99.9±0.1 | 100.0±0.0 |
| OPT-IML-Max-30B | 87.2±1.5 | 74.2±1.7 | 43.5±13.1 | 37.5±9.7 | 64.1±1.8 | 79.6±1.7 | 11.3±14.8 | 42.8±30.4 |
| OPT-IML-Max-1.3B | 73.9±3.4 | 71.2±6.1 | 46.2±0.0 | 46.2±0.0 | 67.9±5.5 | 82.6±6.1 | 100.0±0.0 | 100.0±0.0 |
| OPT-IML-30B | 89.3±1.0 | 75.1±3.8 | 20.4±15.4 | 42.0±3.3 | 53.7±2.3 | 77.6±4.4 | 39.1±20.0 | 95.8±3.3 |
| OPT-IML-1.3B | 77.1±4.7 | 78.5±6.7 | 46.2±0.0 | 46.0±0.3 | 64.2±6.7 | 74.8±6.9 | 100.0±0.0 | 99.8±0.3 |
| BLOOM | 100.0±0.1 | 94.1±1.7 | 53.8±0.0 | 46.0±0.3 | 53.0±0.1 | 59.3±1.9 | 0.0±0.0 | 99.8±0.3 |
| BLOOM-7.1B | 72.3±0.8 | 75.5±3.3 | 53.8±0.2 | 47.0±2.4 | 61.3±4.1 | 61.9±10.0 | 0.2±0.4 | 90.0±31.6 |
| BLOOM-3B | 53.2±0.1 | 53.8±0.0 | 53.8±0.0 | 46.2±0.0 | 99.8±0.1 | 100.0±0.0 | 0.0±0.0 | 100.0±0.0 |
| BLOOM-1.7B | 52.7±3.3 | 46.3±0.2 | 46.2±0.0 | 46.2±0.0 | 9.8±7.1 | 0.1±0.4 | 100.0±0.0 | 100.0±0.0 |
| BLOOMZ | 92.0±1.8 | 55.7±0.6 | 45.0±0.6 | 46.2±0.1 | 60.6±2.0 | 98.1±0.6 | 98.8±0.6 | 99.9±0.2 |
| BLOOMZ-7.1B | 57.3±1.2 | 54.4±0.5 | 46.2±0.0 | 46.2±0.0 | 95.6±1.3 | 99.4±0.5 | 100.0±0.0 | 100.0±0.0 |
| BLOOMZ-3B | 54.4±0.3 | 54.1±0.3 | 46.2±0.0 | 46.2±0.0 | 98.4±0.4 | 98.8±0.4 | 100.0±0.0 | 100.0±0.0 |
| BLOOMZ-1.7B | 57.6±2.1 | 53.8±0.0 | 46.2±0.0 | 46.2±0.0 | 94.4±2.7 | 100.0±0.0 | 100.0±0.0 | 100.0±0.0 |
| NeoXT-Chat-Base-20B | 66.9±6.2 | 55.8±2.2 | 46.2±0.0 | 46.2±0.0 | 86.1±6.2 | 98.0±2.2 | 100.0±0.0 | 100.0±0.0 |

Table 10: Average and standard deviation of model accuracies and the no-ratio of the model outputs at each setting for the **OCCUPATION TASK**.

| Model | Accuracy | | | | No-ratio | | | |
|---|---|---|---|---|---|---|---|---|
| | BASE | FIC | FICNEG | FICNEG-O | BASE | FIC | FICNEG | FICNEG-O |
| GPT-4 | 100.0±0.0 | 88.6±13.6 | 98.1±5.3 | 77.8±12.1 | 54.0±0.0 | 65.0±1.4 | 44.1±0.5 | 32.8±1.9 |
| GPT-3.5-turbo | 73.5±4.5 | 68.1±10.0 | 63.6±8.5 | 35.3±10.2 | 80.3±0.5 | 70.9±1.2 | 73.2±0.9 | 70.9±2.0 |
| text-davinci-003 | 99.3±2.2 | 95.8±4.5 | 94.2±5.4 | 16.2±12.7 | 53.3±0.2 | 49.8±0.4 | 44.8±0.6 | 47.4±1.9 |
| text-davinci-002 | 96.3±2.9 | 88.8±8.7 | 99.3±0.5 | 46.0±0.0 | 57.7±0.3 | 65.2±0.9 | 45.3±0.0 | 100.0±0.0 |
| LLaMA-65B | 100.0±0.0 | 100.0±0.0 | 66.0±10.1 | 43.6±3.8 | 54.0±0.0 | 54.0±0.0 | 80.0±1.0 | 97.6±0.4 |
| LLaMA-30B | 99.3±0.8 | 89.3±5.3 | 88.0±8.2 | 44.3±1.5 | 53.3±0.1 | 43.3±0.5 | 58.0±0.8 | 98.3±0.1 |
| LLaMA-13B | 83.7±5.8 | 91.4±6.3 | 82.2±8.9 | 46.0±0.0 | 70.3±0.6 | 62.4±0.6 | 62.8±1.0 | 100.0±0.0 |
| LLaMA-7B | 68.2±5.8 | 60.2±4.1 | 57.2±11.6 | 46.0±0.0 | 85.8±0.6 | 93.8±0.4 | 88.6±1.2 | 100.0±0.0 |
| Vicuna-13B | 70.7±4.0 | 84.8±5.2 | 93.4±2.6 | 40.6±12.4 | 83.3±0.4 | 69.2±0.5 | 42.0±0.4 | 88.6±0.9 |
| Vicuna-7B | 90.1±3.4 | 89.0±4.4 | 79.9±3.9 | 43.6±5.7 | 62.7±0.4 | 45.8±0.5 | 44.3±0.7 | 97.6±0.6 |
| Alpaca-7B | 72.1±7.6 | 62.9±4.9 | 46.0±0.0 | 46.0±0.0 | 81.9±0.8 | 90.7±0.5 | 100.0±0.0 | 100.0±0.0 |
| OPT-175B | 80.4±15.2 | 58.0±11.6 | 53.2±7.6 | 41.6±12.9 | 34.4±1.5 | 12.0±1.2 | 33.2±4.2 | 95.6±1.3 |
| OPT-66B | 86.3±6.6 | 69.9±5.5 | 43.2±2.7 | 46.0±0.0 | 64.3±0.9 | 83.7±0.6 | 91.8±1.4 | 100.0±0.0 |
| OPT-30B | 54.0±0.0 | 54.0±0.0 | 44.7±3.7 | 46.0±0.0 | 100.0±0.0 | 100.0±0.0 | 44.3±2.6 | 100.0±0.0 |
| OPT-13B | 77.3±11.0 | 78.3±8.1 | 46.0±0.0 | 46.0±0.0 | 34.1±1.4 | 45.1±1.9 | 100.0±0.0 | 100.0±0.0 |
| OPT-6.7B | 55.1±1.1 | 54.2±0.4 | 46.0±0.0 | 46.0±0.0 | 98.9±0.1 | 99.8±0.0 | 100.0±0.0 | 100.0±0.0 |
| OPT-2.7B | 65.8±6.7 | 71.5±7.9 | 53.8±5.1 | 46.0±0.0 | 26.8±1.4 | 52.9±2.0 | 63.0±3.5 | 100.0±0.0 |
| OPT-1.3B | 62.0±6.4 | 59.1±6.0 | 46.0±0.0 | 46.0±0.0 | 26.0±2.1 | 26.5±1.9 | 100.0±0.0 | 100.0±0.0 |
| OPT-IML-Max-30B | 97.7±0.7 | 93.3±3.2 | 25.2±18.3 | 33.2±10.3 | 56.1±0.1 | 60.7±0.3 | 31.2±1.8 | 71.2±1.8 |
| OPT-IML-Max-1.3B | 85.2±8.0 | 80.0±10.4 | 46.0±0.0 | 46.0±0.0 | 53.8±1.5 | 71.2±1.3 | 100.0±0.0 | 100.0±0.0 |
| OPT-IML-30B | 92.5±3.5 | 99.2±1.0 | 28.1±10.4 | 38.0±10.9 | 46.5±0.4 | 53.6±0.1 | 30.5±1.4 | 90.0±1.1 |
| OPT-IML-1.3B | 46.4±1.3 | 48.3±7.3 | 39.8±10.3 | 40.7±10.3 | 0.6±0.2 | 2.3±0.7 | 93.2±1.2 | 90.9±2.0 |
| BLOOM | 87.6±6.8 | 83.7±8.1 | 50.9±6.8 | 45.4±1.6 | 64.2±0.9 | 41.7±1.1 | 93.9±1.0 | 98.2±0.5 |
| BLOOM-7.1B | 46.0±0.0 | 46.0±0.0 | 54.0±0.0 | 47.4±3.6 | 0.0±0.0 | 0.0±0.0 | 0.6±0.2 | 69.8±4.6 |
| BLOOM-3B | 54.0±0.0 | 54.0±0.0 | 54.0±0.0 | 46.0±0.0 | 100.0±0.0 | 100.0±0.0 | 0.0±0.0 | 100.0±0.0 |
| BLOOM-1.7B | 46.0±0.0 | 46.0±0.0 | 46.0±0.0 | 46.0±0.0 | 0.0±0.0 | 0.0±0.0 | 100.0±0.0 | 100.0±0.0 |
| BLOOMZ | 54.1±0.3 | 54.0±0.0 | 46.0±0.0 | 46.4±1.0 | 99.9±0.0 | 100.0±0.0 | 100.0±0.0 | 92.4±1.7 |
| BLOOMZ-7.1B | 54.0±0.0 | 54.0±0.0 | 46.0±0.0 | 46.0±0.0 | 100.0±0.0 | 100.0±0.0 | 100.0±0.0 | 100.0±0.0 |
| BLOOMZ-3B | 59.8±3.0 | 54.5±0.5 | 46.0±0.0 | 46.0±0.0 | 94.2±0.3 | 99.5±0.1 | 100.0±0.0 | 100.0±0.0 |
| BLOOMZ-1.7B | 54.0±0.0 | 54.0±0.0 | 46.0±0.0 | 46.0±0.0 | 100.0±0.0 | 100.0±0.0 | 100.0±0.0 | 100.0±0.0 |
| NeoXT-Chat-Base-20B | 89.1±8.3 | 68.0±11.4 | 46.0±0.0 | 46.2±0.6 | 64.5±0.9 | 84.6±1.4 | 100.0±0.0 | 99.8±0.1 |

Table 11: Average and standard deviation of model accuracies and the no-ratio of the model outputs at each setting for the WEIGHT TRANS. TASK.

| Model | Accuracy | | | | No-ratio | | | |
|---|---|---|---|---|---|---|---|---|
| | BASE | FIC | FICNEG | FICNEG-O | BASE | FIC | FICNEG | FICNEG-O |
| GPT-4 | 92.8±7.0 | 66.9±15.6 | 94.1±2.1 | 82.8±18.1 | 48.7±0.3 | 80.0±1.7 | 47.6±0.3 | 46.9±0.9 |
| GPT-3.5-turbo | 99.5±0.7 | 59.9±3.6 | 71.4±10.8 | 40.0±5.3 | 54.0±0.5 | 88.6±0.5 | 22.9±1.2 | 87.5±0.6 |
| LLaMA-65B | 100.0±0.0 | 100.0±0.0 | 100.0±0.0 | 6.6±6.5 | 53.5±0.5 | 48.5±0.2 | 51.5±0.2 | 55.1±0.8 |
| LLaMA-30B | 100.0±0.0 | 100.0±0.0 | 97.2±3.5 | 23.5±16.8 | 53.5±0.5 | 48.5±0.2 | 48.7±0.5 | 36.7±1.4 |
| LLaMA-13B | 100.0±0.0 | 98.0±2.6 | 99.2±1.3 | 16.1±15.1 | 53.5±0.5 | 50.5±0.4 | 50.8±0.1 | 64.5±1.6 |
| LLaMA-7B | 91.5±8.9 | 88.3±11.7 | 84.6±8.1 | 46.5±5.5 | 62.0±1.3 | 60.2±1.3 | 36.2±0.9 | 95.0±0.7 |
| Vicuna-13B | 100.0±0.0 | 90.7±7.1 | 95.2±4.2 | 19.6±10.6 | 53.5±0.5 | 57.8±0.9 | 46.7±0.5 | 53.0±1.6 |
| OPT-175B | 98.5±1.8 | 100.0±0.0 | 91.8±6.9 | 34.2±17.8 | 55.1±0.6 | 48.5±0.2 | 48.8±0.6 | 82.7±1.9 |
| OPT-66B | 96.5±3.9 | 100.0±0.0 | 25.8±11.8 | 13.9±16.4 | 57.0±0.8 | 48.5±0.2 | 28.9±1.4 | 62.4±1.7 |
| OPT-30B | 89.5±7.2 | 74.5±13.9 | 49.9±2.8 | 51.5±1.5 | 64.0±1.2 | 74.0±1.5 | 97.1±0.1 | 100.0±0.0 |
| OPT-13B | 83.5±10.6 | 83.9±15.2 | 63.4±20.0 | 51.5±1.5 | 70.0±1.5 | 64.5±1.7 | 67.3±1.8 | 100.0±0.0 |
| OPT-6.7B | 93.7±6.5 | 99.0±1.0 | 50.1±3.4 | 41.6±14.4 | 59.8±1.1 | 49.5±0.3 | 98.6±0.2 | 90.2±1.5 |
| BLOOM | 100.0±0.0 | 99.4±0.9 | 51.7±5.1 | 43.6±8.1 | 53.5±0.5 | 49.1±0.1 | 3.3±0.4 | 92.1±0.8 |
| BLOOMZ | 96.0±1.3 | 50.3±2.3 | 51.2±3.2 | 64.0±12.6 | 57.5±0.5 | 98.2±0.1 | 95.9±0.4 | 86.9±1.4 |

Table 12: Average and standard deviation of model accuracies and the *no*-ratio at each setting for different **Prompt format**.

| Model | Accuracy | | | | No-ratio | | | |
|---|---|---|---|---|---|---|---|---|
| | BASE | FIC | FICNEG | FICNEG-O | BASE | FIC | FICNEG | FICNEG-O |
| GPT-4 (yes-no-yes) | 98.1±1.3 | 64.2±13.8 | 90.9±5.0 | 69.8±15.5 | 54.3±0.5 | 84.2±1.3 | 42.4±0.5 | 63.0±1.6 |
| GPT-4 (no-yes-yes) | 98.0±1.0 | 70.0±15.6 | 87.9±5.2 | 66.6±19.2 | 54.5±0.4 | 78.4±1.4 | 39.4±0.5 | 68.4±1.6 |
| GPT-4 (yes-yes-no) | 98.0±1.2 | 71.0±14.1 | 96.4±2.2 | 72.0±17.6 | 54.2±0.4 | 77.5±1.3 | 49.6±0.4 | 51.1±1.7 |
| GPT-3.5-turbo (yes-no-yes) | 98.6±1.7 | 55.1±2.8 | 66.4±8.4 | 41.9±4.4 | 54.6±0.6 | 93.4±0.4 | 17.9±1.0 | 90.4±0.6 |
| GPT-3.5-turbo (no-yes-yes) | 98.0±2.2 | 54.9±3.5 | 71.0±7.1 | 41.4±4.3 | 55.5±0.7 | 93.5±0.5 | 22.8±0.9 | 90.0±0.6 |
| GPT-3.5-turbo (yes-yes-no) | 99.2±0.8 | 60.4±5.1 | 75.1±7.5 | 34.7±8.2 | 53.1±0.4 | 88.1±0.6 | 34.9±0.9 | 83.2±0.9 |
| LLaMA-65B (yes-no-yes) | 100.0±0.0 | 95.3±5.4 | 96.1±4.7 | 21.7±13.7 | 53.5±0.5 | 43.8±0.7 | 55.3±0.6 | 70.2±1.3 |
| LLaMA-65B (no-yes-yes) | 100.0±0.0 | 91.1±4.0 | 91.2±4.1 | 39.3±9.4 | 53.5±0.5 | 51.9±1.1 | 52.0±1.1 | 87.8±1.0 |
| LLaMA-65B (yes-yes-no) | 99.7±0.5 | 90.1±10.8 | 90.1±9.8 | 9.2±12.1 | 53.2±0.5 | 38.6±1.2 | 61.4±1.1 | 56.2±1.2 |
| LLaMA-30B (yes-no-yes) | 100.0±0.0 | 94.5±6.0 | 93.7±6.0 | 14.3±14.0 | 53.5±0.5 | 43.0±0.7 | 45.7±0.8 | 56.8±1.4 |
| LLaMA-30B (no-yes-yes) | 97.2±2.0 | 93.7±5.2 | 89.1±5.1 | 14.1±15.2 | 56.2±0.5 | 44.7±0.7 | 42.9±0.9 | 47.3±1.6 |
| LLaMA-30B (yes-yes-no) | 98.6±0.5 | 83.7±16.8 | 87.8±11.6 | 21.9±17.2 | 52.1±0.5 | 32.2±1.8 | 61.3±1.5 | 29.6±1.7 |
| LLaMA-13B (yes-no-yes) | 99.0±1.3 | 74.3±7.2 | 70.7±19.2 | 36.8±13.5 | 53.3±0.5 | 66.4±1.6 | 63.6±2.3 | 83.7±1.7 |
| LLaMA-13B (no-yes-yes) | 91.2±3.2 | 54.0±3.9 | 51.9±3.4 | 48.4±4.6 | 62.3±0.5 | 93.2±0.6 | 95.0±0.8 | 96.8±0.5 |
| LLaMA-13B (yes-yes-no) | 98.5±1.5 | 86.6±7.2 | 71.2±17.3 | 27.9±17.3 | 52.0±0.6 | 54.3±1.5 | 73.4±2.2 | 69.0±2.7 |
| LLaMA-7B (yes-no-yes) | 82.3±12.9 | 58.0±5.4 | 54.8±6.4 | 51.5±1.5 | 71.2±1.8 | 90.5±0.7 | 7.5±0.8 | 100.0±0.0 |
| LLaMA-7B (no-yes-yes) | 74.9±6.9 | 50.5±2.7 | 39.0±9.4 | 51.5±1.5 | 78.6±1.1 | 98.0±0.2 | 77.9±2.1 | 100.0±0.0 |
| LLaMA-7B (yes-yes-no) | 96.3±2.2 | 82.6±7.2 | 40.2±9.5 | 50.9±2.2 | 54.9±0.9 | 65.9±0.8 | 79.2±1.6 | 99.4±0.2 |
| Vicuna-13B (yes-no-yes) | 98.7±0.9 | 75.1±8.0 | 78.5±10.1 | 24.0±8.2 | 52.8±0.6 | 62.2±2.1 | 36.4±1.8 | 57.7±2.2 |
| Vicuna-13B (no-yes-yes) | 96.0±2.2 | 68.2±13.9 | 66.3±12.0 | 29.7±10.2 | 56.6±0.7 | 78.5±1.7 | 32.3±2.8 | 68.3±2.1 |
| Vicuna-13B (yes-yes-no) | 98.8±1.3 | 79.9±6.3 | 79.7±8.3 | 27.8±8.3 | 52.2±0.6 | 58.4±1.8 | 33.9±1.3 | 59.5±2.2 |
| OPT-175B (yes-no-yes) | 95.5±2.9 | 80.0±15.6 | 41.2±14.0 | 49.0±8.0 | 52.6±0.7 | 30.2±1.9 | 20.5±1.9 | 97.5±0.8 |
| OPT-175B (no-yes-yes) | 91.2±8.1 | 90.6±4.1 | 41.4±11.0 | 49.2±7.6 | 56.6±0.9 | 50.5±1.1 | 10.7±1.5 | 97.8±0.8 |
| OPT-175B (yes-yes-no) | 93.3±1.7 | 87.7±13.4 | 37.1±15.1 | 44.4±12.8 | 47.4±0.6 | 36.2±1.5 | 29.5±3.1 | 92.9±1.3 |
| OPT-66B (yes-no-yes) | 90.8±3.2 | 93.3±6.6 | 33.7±8.2 | 40.2±11.8 | 59.9±1.1 | 44.9±0.7 | 19.0±1.2 | 88.7±1.2 |
| OPT-66B (no-yes-yes) | 83.3±8.1 | 84.8±10.7 | 20.2±10.6 | 50.7±1.8 | 67.4±1.6 | 61.7±1.5 | 54.1±1.8 | 99.2±0.2 |
| OPT-66B (yes-yes-no) | 86.8±5.6 | 89.9±6.0 | 18.9±14.8 | 47.9±6.6 | 65.7±1.1 | 58.4±0.7 | 58.8±1.7 | 96.4±0.7 |
| OPT-30B (yes-no-yes) | 68.9±6.8 | 48.9±1.3 | 46.0±5.4 | 51.5±1.5 | 84.6±1.1 | 99.6±0.1 | 93.1±0.7 | 100.0±0.0 |
| OPT-30B (no-yes-yes) | 61.1±5.5 | 48.8±1.9 | 50.9±2.2 | 51.5±1.5 | 92.5±0.1 | 99.7±0.1 | 99.5±0.1 | 100.0±0.0 |
| OPT-30B (yes-yes-no) | 57.1±2.6 | 48.5±1.5 | 38.6±14.6 | 51.5±1.5 | 96.4±0.3 | 100.0±0.0 | 85.0±1.6 | 100.0±0.0 |
| OPT-13B (yes-no-yes) | 70.0±6.7 | 66.2±12.8 | 44.2±12.6 | 51.5±1.5 | 75.0±2.0 | 65.8±3.1 | 56.3±2.4 | 100.0±0.0 |
| OPT-13B (no-yes-yes) | 73.8±3.4 | 77.0±5.1 | 51.2±1.4 | 48.7±7.3 | 70.0±1.7 | 47.6±1.8 | 99.7±0.1 | 97.0±0.8 |
| OPT-13B (yes-yes-no) | 71.2±7.7 | 63.8±14.1 | 29.3±11.0 | 51.5±1.5 | 74.2±2.0 | 82.8±1.8 | 68.5±1.5 | 100.0±0.0 |
| OPT-6.7B (yes-no-yes) | 81.0±6.3 | 80.9±4.4 | 40.2±12.7 | 49.5±4.9 | 66.8±1.4 | 49.6±2.0 | 78.9±2.4 | 98.0±0.5 |
| OPT-6.7B (no-yes-yes) | 79.2±8.5 | 67.7±12.2 | 50.9±2.4 | 46.1±10.7 | 50.6±0.8 | 18.2±1.6 | 99.4±0.1 | 93.8±1.4 |
| OPT-6.7B (yes-yes-no) | 68.2±6.2 | 62.3±7.6 | 32.1±7.5 | 51.5±1.5 | 84.2±0.3 | 86.2±0.9 | 45.2±2.6 | 100.0±0.0 |
| BLOOM (yes-no-yes) | 99.2±0.9 | 87.8±3.1 | 48.5±1.5 | 51.5±1.5 | 54.2±0.4 | 60.7±0.3 | 0.0±0.0 | 100.0±0.0 |
| BLOOM (no-yes-yes) | 96.9±1.8 | 75.8±4.0 | 46.5±2.6 | 47.6±5.5 | 56.6±0.6 | 72.8±0.5 | 2.0±0.3 | 89.9±1.7 |
| BLOOM (yes-yes-no) | 99.1±1.0 | 91.0±5.1 | 48.3±1.4 | 49.1±3.4 | 52.6±0.6 | 57.6±0.4 | 0.6±0.1 | 96.2±0.7 |
| BLOOMZ (yes-no-yes) | 90.1±3.2 | 49.1±2.2 | 50.8±1.9 | 51.2±1.5 | 62.6±0.3 | 99.4±0.1 | 99.4±0.0 | 99.6±0.1 |
| BLOOMZ (no-yes-yes) | 86.9±2.5 | 49.0±2.1 | 50.8±1.8 | 51.4±1.5 | 66.2±0.3 | 99.5±0.1 | 99.2±0.0 | 99.9±0.0 |
| BLOOMZ (yes-yes-no) | 87.3±1.9 | 49.1±2.1 | 49.1±1.3 | 48.8±3.4 | 65.6±0.6 | 99.5±0.1 | 97.6±0.2 | 94.2±0.8 |

Table 13: Average and standard deviation of model accuracies and the *no*-ratio of the model outputs at each setting for different **Prompt orders**.

| Model | SPORTS Task | | | | OCCUPATION Task | | | | WEIGHT TRANS. Task | | | |
|---|---|---|---|---|---|---|---|---|---|---|---|---|
| | BASE | FIC | FICNEG | FICNEG-O | BASE | FIC | FICNEG | FICNEG-O | BASE | FIC | FICNEG | FICNEG-O |
| GPT-4 | 0.99±0.0 | 0.46±0.1 | 0.92±0.0 | 0.63±0.2 | 0.98±0.0 | 0.73±0.1 | 0.38±0.4 | 0.76±0.1 | 1.0±0.0 | 0.87±0.2 | 0.98±0.1 | 0.75±0.2 |
| GPT-3.5-turbo | 1.0±0.0 | 0.52±0.0 | 0.7±0.1 | 0.27±0.0 | 0.97±0.0 | 0.46±0.0 | 0.52±0.2 | 0.29±0.0 | 0.7±0.1 | 0.65±0.1 | 0.62±0.1 | 0.31±0.1 |
| text-davinci-003 | 1.0±0.0 | 0.81±0.0 | 0.87±0.0 | 0.26±0.1 | 1.0±0.0 | 0.7±0.0 | 0.49±0.1 | 0.21±0.0 | 0.99±0.0 | 0.96±0.0 | 0.94±0.1 | 0.13±0.1 |
| text-davinci-002 | 1.0±0.0 | 0.73±0.0 | 0.91±0.0 | 0.33±0.0 | 1.0±0.0 | 0.55±0.1 | 0.53±0.1 | 0.38±0.1 | 0.96±0.0 | 0.88±0.1 | 0.99±0.0 | 0.32±0.0 |
| LLaMA-65B | 1.0±0.0 | 0.89±0.0 | 0.91±0.0 | 0.17±0.1 | 1.0±0.0 | 1.0±0.0 | 1.0±0.0 | 0.13±0.0 | 1.0±0.0 | 1.0±0.0 | 0.63±0.1 | 0.31±0.0 |
| LLaMA-30B | 1.0±0.0 | 0.85±0.0 | 0.99±0.0 | 0.05±0.1 | 1.0±0.0 | 1.0±0.0 | 0.87±0.0 | 0.16±0.1 | 0.99±0.0 | 0.89±0.1 | 0.88±0.1 | 0.31±0.0 |
| LLaMA-13B | 0.99±0.0 | 0.77±0.0 | 0.38±0.0 | 0.2±0.1 | 1.0±0.0 | 0.67±0.1 | 0.91±0.0 | 0.3±0.0 | 0.82±0.1 | 0.91±0.1 | 0.82±0.1 | 0.32±0.0 |
| LLaMA-7B | 0.94±0.0 | 0.57±0.1 | 0.5±0.1 | 0.33±0.0 | 0.63±0.0 | 0.48±0.0 | 0.37±0.0 | 0.32±0.0 | 0.62±0.1 | 0.48±0.1 | 0.49±0.2 | 0.32±0.0 |
| Vicuna-13B | 0.98±0.0 | 0.77±0.0 | 0.83±0.0 | 0.19±0.0 | 1.0±0.0 | 0.68±0.0 | 0.7±0.1 | 0.2±0.0 | 0.66±0.1 | 0.84±0.1 | 0.93±0.0 | 0.32±0.1 |
| Vicuna-7B | 0.98±0.0 | 0.93±0.0 | 0.49±0.0 | 0.26±0.2 | 0.93±0.0 | 0.56±0.0 | 0.35±0.0 | 0.28±0.0 | 0.9±0.0 | 0.89±0.0 | 0.8±0.0 | 0.3±0.0 |
| Alpaca-7B | 0.91±0.0 | 0.83±0.1 | 0.41±0.0 | 0.34±0.0 | 0.82±0.0 | 0.82±0.1 | 0.35±0.1 | 0.3±0.0 | 0.67±0.1 | 0.53±0.1 | 0.32±0.0 | 0.32±0.0 |
| OPT-175B | 0.96±0.0 | 0.52±0.1 | 0.55±0.2 | 0.3±0.1 | 0.93±0.0 | 0.92±0.0 | 0.28±0.2 | 0.31±0.0 | 0.79±0.2 | 0.51±0.2 | 0.4±0.1 | 0.29±0.1 |
| OPT-66B | 0.91±0.0 | 0.85±0.0 | 0.31±0.0 | 0.27±0.1 | 0.88±0.0 | 1.0±0.0 | 0.27±0.1 | 0.26±0.1 | 0.85±0.1 | 0.64±0.1 | 0.32±0.0 | 0.32±0.0 |
| OPT-30B | 0.71±0.0 | 0.35±0.0 | 0.36±0.0 | 0.33±0.0 | 0.49±0.0 | 0.36±0.0 | 0.29±0.0 | 0.32±0.0 | 0.35±0.0 | 0.35±0.0 | 0.41±0.0 | 0.32±0.0 |
| OPT-13B | 0.73±0.0 | 0.71±0.1 | 0.45±0.0 | 0.33±0.0 | 0.52±0.0 | 0.46±0.1 | 0.27±0.1 | 0.32±0.0 | 0.76±0.1 | 0.77±0.1 | 0.32±0.0 | 0.32±0.0 |
| OPT-6.7B | 0.86±0.0 | 0.75±0.1 | 0.32±0.0 | 0.31±0.0 | 0.68±0.0 | 0.86±0.0 | 0.24±0.0 | 0.32±0.0 | 0.37±0.0 | 0.35±0.0 | 0.32±0.0 | 0.32±0.0 |
| OPT-2.7B | 0.75±0.1 | 0.44±0.1 | 0.31±0.0 | 0.3±0.0 | 0.36±0.0 | 0.38±0.0 | 0.32±0.0 | 0.32±0.0 | 0.64±0.1 | 0.69±0.1 | 0.47±0.1 | 0.32±0.0 |
| OPT-1.3B | 0.68±0.1 | 0.51±0.1 | 0.33±0.0 | 0.33±0.0 | 0.44±0.0 | 0.6±0.1 | 0.32±0.0 | 0.32±0.0 | 0.58±0.1 | 0.56±0.1 | 0.32±0.0 | 0.32±0.0 |
| OPT-IML-Max-30B | 0.96±0.0 | 0.8±0.1 | 0.41±0.1 | 0.42±0.1 | 0.87±0.0 | 0.71±0.0 | 0.3±0.1 | 0.31±0.1 | 0.98±0.0 | 0.93±0.0 | 0.2±0.1 | 0.29±0.1 |
| OPT-IML-Max-1.3B | 0.48±0.0 | 0.36±0.0 | 0.33±0.0 | 0.33±0.0 | 0.73±0.0 | 0.66±0.1 | 0.32±0.0 | 0.32±0.0 | 0.84±0.1 | 0.77±0.2 | 0.32±0.0 | 0.32±0.0 |
| OPT-IML-30B | 0.95±0.0 | 0.83±0.1 | 0.51±0.1 | 0.35±0.0 | 0.89±0.0 | 0.72±0.0 | 0.16±0.1 | 0.3±0.0 | 0.92±0.0 | 0.99±0.0 | 0.23±0.1 | 0.29±0.1 |
| OPT-IML-1.3B | 0.5±0.0 | 0.39±0.0 | 0.33±0.0 | 0.33±0.0 | 0.76±0.1 | 0.76±0.1 | 0.32±0.0 | 0.32±0.0 | 0.33±0.0 | 0.35±0.1 | 0.28±0.1 | 0.29±0.1 |
| BLOOM | 0.99±0.0 | 0.89±0.0 | 0.34±0.0 | 0.33±0.0 | 1.0±0.0 | 0.94±0.0 | 0.35±0.0 | 0.32±0.0 | 0.87±0.1 | 0.83±0.1 | 0.41±0.1 | 0.32±0.0 |
| BLOOM-7.1B | 0.68±0.0 | 0.37±0.0 | 0.34±0.0 | 0.31±0.0 | 0.72±0.0 | 0.75±0.0 | 0.35±0.0 | 0.35±0.0 | 0.32±0.0 | 0.32±0.0 | 0.35±0.0 | 0.33±0.0 |
| BLOOM-3B | 0.36±0.0 | 0.34±0.0 | 0.37±0.0 | 0.33±0.0 | 0.35±0.0 | 0.35±0.0 | 0.35±0.0 | 0.32±0.0 | 0.35±0.0 | 0.35±0.0 | 0.35±0.0 | 0.32±0.0 |
| BLOOM-1.7B | 0.36±0.0 | 0.33±0.0 | 0.33±0.0 | 0.33±0.0 | 0.45±0.1 | 0.32±0.0 | 0.32±0.0 | 0.32±0.0 | 0.32±0.0 | 0.32±0.0 | 0.32±0.0 | 0.32±0.0 |
| BLOOMZ | 0.91±0.0 | 0.34±0.0 | 0.33±0.0 | 0.33±0.0 | 0.92±0.0 | 0.39±0.0 | 0.31±0.0 | 0.32±0.0 | 0.35±0.0 | 0.35±0.0 | 0.32±0.0 | 0.35±0.0 |
| BLOOMZ-7.1B | 0.39±0.0 | 0.34±0.0 | 0.33±0.0 | 0.33±0.0 | 0.44±0.0 | 0.36±0.0 | 0.32±0.0 | 0.32±0.0 | 0.35±0.0 | 0.35±0.0 | 0.32±0.0 | 0.32±0.0 |
| BLOOMZ-3B | 0.34±0.0 | 0.34±0.0 | 0.33±0.0 | 0.33±0.0 | 0.38±0.0 | 0.37±0.0 | 0.32±0.0 | 0.32±0.0 | 0.47±0.1 | 0.36±0.0 | 0.32±0.0 | 0.32±0.0 |
| BLOOMZ-1.7B | 0.34±0.0 | 0.34±0.0 | 0.33±0.0 | 0.33±0.0 | 0.45±0.0 | 0.35±0.0 | 0.32±0.0 | 0.32±0.0 | 0.35±0.0 | 0.35±0.0 | 0.32±0.0 | 0.32±0.0 |
| NeoXT-Chat-Base-20B | 0.76±0.0 | 0.38±0.0 | 0.33±0.0 | 0.33±0.0 | 0.6±0.1 | 0.39±0.0 | 0.32±0.0 | 0.32±0.0 | 0.88±0.1 | 0.6±0.2 | 0.32±0.0 | 0.32±0.0 |

Table 14: Models' average-macro F1 score at each settings.

| Model Name | Details | URL |
| --- | --- | --- |
| GPT-4 | OpenAI's API call, specifically gpt-4-0314 | https://platform.openai.com/docs/api-reference/chat |
| GPT-3.5-turbo | OpenAI's API call, specifically gpt-3.5-turbo-0301 | https://platform.openai.com/docs/api-reference/chat |
| text-davinci-003 | OpenAI's API call, specifically text-davinci-003 | https://platform.openai.com/docs/api-reference/completions |
| text-davinci-002 | OpenAI's API call, specifically text-davinci-002 | https://platform.openai.com/docs/api-reference/completions |
| LLaMA-65B | Original model weight provided by LLAMA Release Team | https://github.com/facebookresearch/llama |
| LLaMA-30B | Original model weight provided by LLAMA Release Team | https://github.com/facebookresearch/llama |
| LLaMA-13B | Original model weight provided by LLAMA Release Team | https://github.com/facebookresearch/llama |
| LLaMA-7B | Original model weight provided by LLAMA Release Team | https://github.com/facebookresearch/llama |
| Vicuna-13B | Recovered model weight based on LLaMA | https://huggingface.co/lmsys/vicuna-13b-delta-v1.1 |
| Vicuna-7B | Recovered model weight based on LLaMA | https://huggingface.co/lmsys/vicuna-7b-delta-v1.1 |
| Alpaca-7B | Recovered model weight based on LLaMA | https://huggingface.co/tatsu-lab/alpaca-7b-wdiff |
| OPT-175B | Original model weights provided by Meta | https://github.com/facebookresearch/metaseq/blob/main/projects/OPT/download_opt175b.md |
| OPT-66B | Available on Huggingface | https://huggingface.co/facebook/opt-66b |
| OPT-30B | Available on Huggingface | https://huggingface.co/facebook/opt-30b |
| OPT-13B | Available on Huggingface | https://huggingface.co/facebook/opt-13b |
| OPT-6.7B | Available on Huggingface | https://huggingface.co/facebook/opt-6.7b |
| OPT-2.7B | Available on Huggingface | https://huggingface.co/facebook/opt-2.7b |
| OPT-1.3B | Available on Huggingface | https://huggingface.co/facebook/opt-1.3b |
| OPT-IML-Max-30B | Available on Huggingface | https://huggingface.co/facebook/opt-iml-max-30b |
| OPT-IML-Max-30B | Available on Huggingface | https://huggingface.co/facebook/opt-iml-max-1.3b |
| OPT-IML-30B | Available on Huggingface | https://huggingface.co/facebook/opt-iml-30b |
| OPT-IML-30B | Available on Huggingface | https://huggingface.co/facebook/opt-iml-1.3b |
| BLOOM | Available on Huggingface | https://huggingface.co/bigscience/bloom |
| BLOOM-7.1B | Available on Huggingface | https://huggingface.co/bigscience/bloom-7b1 |
| BLOOM-3B | Available on Huggingface | https://huggingface.co/bigscience/bloom-3b |
| BLOOM-1.7B | Available on Huggingface | https://huggingface.co/bigscience/bloom-1b7 |
| BLOOMZ | Available on Huggingface | https://huggingface.co/bigscience/bloomz |
| BLOOMZ-7.1B | Available on Huggingface | https://huggingface.co/bigscience/bloomz-7b1 |
| BLOOMZ-3B | Available on Huggingface | https://huggingface.co/bigscience/bloomz-3b |
| BLOOMZ-1.7B | Available on Huggingface | https://huggingface.co/bigscience/bloomz-1b7 |
| NeoXT-Chat-Base-20B | Available on Huggingface | https://huggingface.co/togethercomputer/GPT-NeoXT-Chat-Base-20B |

Table 15: Full model list and details.

| Few-shot exemplars |
| --- |
| Determine whether an artificially constructed sentence relating to sports is plausible or not. |
| Q: Is the following sentence plausible? "Bam Adebayo scored a reverse layup in the Western Conference Finals."
A: Let's think step by step.
Bam Adebayo is an American basketball player. Scoring a reverse layup happens in basketball. So the answer is yes.

Q: Is the following sentence plausible? "Santi Cazorla scored a touchdown."
A: Let's think step by step.
Santi Cazorla is a soccer player. Touchdown happens in football. So the answer is no.

Q: Is the following sentence plausible? "DeMar DeRozan was called for the goaltend."
A: Let's think step by step.
DeMar DeRozan is an American basketball player. Goaltending happens in basketball. So the answer is yes. |
| Target example |
| Q: Is the following sentence plausible? "Raisel Iglesias was safe at first."
A: Let's think step by step.
Raisel Iglesias is a baseball player. Getting out at first happens in baseball. So the answer is __ |

Table 16: Example of model input for BASE setting, i.e., entity choices are realistic, neither few-shot exemplars nor the target example has lexical negation.

| Few-shot exemplars |
| --- |
| Determine whether an artificially constructed sentence relating to fiction sports is plausible or not. |
| Q: Is the following sentence plausible? "Bam Adebayo scored a reverse layup in the Western Conference Finals."
A: Let's think step by step.
Bam Adebayo is an American basketball player. Scoring a reverse layup happens in basketball. So the answer is yes.

Q: Is the following sentence plausible? "Santi Cazorla scored a touchdown."
A: Let's think step by step.
Santi Cazorla is a soccer player. Touchdown happens in football. So the answer is no.

Q: Is the following sentence plausible? "DeMar DeRozan was called for the goaltend."
A: Let's think step by step.
DeMar DeRozan is an American basketball player. Goaltending happens in basketball. So the answer is yes. |
| Target example |
| Q: Is the following sentence plausible? "Harrison Bullock was safe at first."
A: Let's think step by step.
Harrison Bullock is a turboglide player. Getting out at first happens in turboglide. So the answer is __ |

Table 17: Example of model input for fictional setting (FIC), i.e., entity choices are fictional, neither few-shot exemplars nor the target example has lexical negation.

| Few-shot exemplars |
|---|
| Determine whether an artificially constructed sentence relating to fiction sports is implausible or not. |
| Q: Is the following sentence implausible? "Bam Adebayo scored a reverse layup in the Western Conference Finals." A: Let's think step by step. Bam Adebayo is an American basketball player. Scoring a reverse layup happens in basketball. So the answer is no. |
| Q: Is the following sentence implausible? "Santi Cazorla scored a touchdown." A: Let's think step by step. Santi Cazorla is a soccer player. Touchdown happens in football. So the answer is yes. |
| Q: Is the following sentence implausible? "DeMar DeRozan was called for the goaltend." A: Let's think step by step. DeMar DeRozan is an American basketball player. Goaltending happens in basketball. So the answer is no. |

| Target example |
|---|
| Q: Is the following sentence implausible? "Harrison Bullock was safe at first." A: Let's think step by step. Harrison Bullock is a turboglide player. Getting out at first happens in turboglide. So the answer is __ |

Table 18: Example of model input for in-domain negation setting (FICNEG), i.e., entity choices are fictional, both few-shot exemplars and the target example have lexical negation.

| Few-shot exemplars |
|---|
| Determine whether an artificially constructed sentence relating to fiction sports is plausible or not. |
| Q: Is the following sentence plausible? "Bam Adebayo scored a reverse layup in the Western Conference Finals." A: Let's think step by step. Bam Adebayo is an American basketball player. Scoring a reverse layup happens in basketball. So the answer is yes. |
| Q: Is the following sentence plausible? "Santi Cazorla scored a touchdown." A: Let's think step by step. Santi Cazorla is a soccer player. Touchdown happens in football. So the answer is no. |
| Q: Is the following sentence plausible? "DeMar DeRozan was called for the goaltend." A: Let's think step by step. DeMar DeRozan is an American basketball player. Goaltending happens in basketball. So the answer is yes. |

| Target example |
|---|
| Q: Is the following sentence implausible? "Harrison Bullock was safe at first." A: Let's think step by step. Harrison Bullock is a turboglide player. Getting out at first happens in turboglide. So the answer is __ |

Table 19: Example of model input for out-domain negation setting (FICNEG-O), i.e., entity choices are fictional, only the target example has lexical negation.