# OpenReview forum: "Assessing Step-by-Step Reasoning against Lexical Negation: A Case Study on Syllogism"
_EMNLP/2023/Conference — EMNLP 2023 Main_

### Official Review · Reviewer_RddC · 2023-08-01

**Soundness:** 4

**Excitement:**

4: Strong: This paper deepens the understanding of some phenomenon or lowers the barriers to an existing research direction.

**Paper Topic And Main Contributions:**

The paper investigates the ability of modern LLMs to perform syllogisms involving morphologically negative words, with Chain of Thoughts systems.

**Reasons To Accept:**

The paper highlights a basic failure of even some of the most apt modern LLMs.

**Reasons To Reject:**

Failures of LLMs at various tasks is likely to happen. It is interesting that the paper here tries to overcome a very general "there is problem" alter by decomposing the task, to understand the exact source of the failure.

**Reproducibility:**

5: Could easily reproduce the results.

**Reviewer Confidence:**

3: Pretty sure, but there's a chance I missed something. Although I have a good feel for this area in general, I did not carefully check the paper's details, e.g., the math, experimental design, or novelty.

---

> ### Author Rebuttal · Authors · 2023-08-29
>
> Thank you for your review.
>
> **Reasons To Reject: Failures of LLMs at various tasks is likely to happen.It is interesting that the paper here tries to overcome this by trying to understand the decompose the task, to understand the exact source of the failure.**
>
> We’re struggling with comprehending the Reasons to Reject you provided fully. Per our understanding, you state that pinpointing the specific source of LLMs' failures by decomposing the task is an interesting direction, and this is exactly what our study does (see Table 1). If this is the case, we view it as a positive comment, and there seems to be a disparity between the soundness score (3) and your reason to reject. The other two reviewers have given positive reviews and scores (soundness: 4 and 5) that align with their positive reviews. Therefore, we kindly request that you reconsider your score so it aligns with your review.

---

### Official Review · Reviewer_a7Sc · 2023-08-05

**Soundness:** 5

**Excitement:**

4: Strong: This paper deepens the understanding of some phenomenon or lowers the barriers to an existing research direction.

**Paper Topic And Main Contributions:**

This paper studies whether large language models are able to correctly answer questions requiring step-by-step reasoning, and whether these abilities are robust in the presence of lexical negation in the prompts. The authors sample instances from the BIG-Bench sports-understanding task (and generate similar instances in the occupation and weight-transitivity domains), and modify them with fictional entities and negated words to create four increasingly difficult tasks. Their results show decreasing performance as the tasks become harder, with different model families beginning to struggle at different points, and no performance improvement (or even a performance decrease) with increasing model size.

**Questions For The Authors:**

Question A: The authors “used ten variants of negated words, e.g., implausible, unreasonable” (lines 119-121). To be clear, were the base (i.e., non-negated) words also used?

**Reasons To Accept:**

This paper is very well-written and easy to read overall. The methods are sound, and the authors conducted an impressive number of auxiliary experiments, on various domains, models, and prompt styles, to ensure the robustness of the results. The results themselves, particularly that different model families break at different points, are very interesting and show the importance of differences in language models beyond parameter size.

**Reasons To Reject:**

The results shown in the main text (i.e., Tables 2 and 3) average together results from both the sports and occupation domains, which can mask differences in model behavior between the different domains. For example, in Tables 8 and 9, it seems that most OPT models in the FIC setting tend to answer “yes” in the sports task and “no” in the occupation task, while one might conclude from Table 3 that they tend to answer with a mix of “yes” and “no”. I would suggest using one domain for Tables 2 and 3 and referring to the Appendix for the others.

**Reproducibility:**

4: Could mostly reproduce the results, but there may be some variation because of sample variance or minor variations in their interpretation of the protocol or method.

**Reviewer Confidence:**

3: Pretty sure, but there's a chance I missed something. Although I have a good feel for this area in general, I did not carefully check the paper's details, e.g., the math, experimental design, or novelty.

**Typos Grammar Style And Presentation Improvements:**

Looking at Tables 2 and 3, the most striking thing I notice is that on the FicNeg-O task, with the notable exception of GPT-4, the best-performing models are precisely those that always answer “no”, and therefore get ~50% accuracy. The authors may want to consider (additionally) presenting metrics other than accuracy; for example, the minimum of the F1 scores between the two classes (or even the macro-averaged F1 score) will sharply penalize those models that always answer “no” (or “yes”).

The authors note in Appendix C that they experiment with a weight transitivity task; however, this is not mentioned in the main text—it should be (not everyone will read the appendix, and the authors should make sure the readers know about all the work they have done!).

---

> ### Author Rebuttal · Authors · 2023-08-29
>
> We appreciate your constructive comments!
>
> **The averaged results in the main text could mask differences in model behavior between the different domains.**
>
> We appreciate your perceptive feedback. We fully agree that averaging behaviors across both sports and occupation domains (Tables 2 and 3) may result in an inadequate comprehension of the varying behaviors between these domains although we separately showed the per-task results in Appendix E. Following your suggestion, we will divide Tables 2 and 3 into two separate tables each and position them suitably in the camera-ready version.
>
> **Question A: Were the non-negated words also used in the experiments?**
>
> Yes, we did utilize all 10 non-negated words (e.g., plausible, reasonable) shown in Table 4 in our experiments, although it wasn't clearly mentioned in the description of our BASE setting (lines 114-116). We'll ensure this is clarified in the camera-ready version.
>
> **Additional metrics other than accuracy would sharply penalize those models that always answer “no” (or “yes”).**
>
> We agree with your suggestion that offering additional metrics like F1 scores would facilitate a better understanding of the various trends between different LLMs. We will incorporate F1 scores and an expanded discussion in the camera-ready version.

---

### Official Review · Reviewer_YrJj · 2023-08-05

**Typos Grammar Style And Presentation Improvements:** The caption of Table 5 in the appendi…
**Soundness:** 4

**Excitement:**

4: Strong: This paper deepens the understanding of some phenomenon or lowers the barriers to an existing research direction.

**Paper Topic And Main Contributions:**

This paper conducts a controlled assessment of large language models' robustness to negation when instructed to evaluate syllogisms involving fictional referents. The authors evaluate 31 models (results for 14 of which are presented in the main text), finding that model performance consistently degrades when fictional referents are used, and when negation in instructions is not matched between demonstrations and the target example. These findings support the conclusion that models lack enough "pure reasoning" ability to faithfully carry out chain-of-thought reasoning in general, and that CoT performance instead relies on a combination of factors beyond just the structure of the reasoning chains.

**Questions For The Authors:**

A: How does performance in the Fic setting change if you provide fictional few-shot exemplars instead of real ones?

**Reasons To Accept:**

The evaluation is thorough and covers a wide range of models, giving readers a full picture beyond just "LLMs fail to do X".

The experiment is well-designed to help disentangle some of the factors contributing to CoT's effectiveness.

**Reasons To Reject:**

I did not notice any methodological flaws that would be cause for rejection.

**Reproducibility:**

5: Could easily reproduce the results.

**Reviewer Confidence:**

4: Quite sure. I tried to check the important points carefully. It's unlikely, though conceivable, that I missed something that should affect my ratings.

---

> ### Author Rebuttal · Authors · 2023-08-29
>
> We thank you for your invaluable review!
>
> **Question A: What would happen if you provided fictional few-shot examples in the FIC setting?**
>
> We appreciate your insightful query. We are willing to share the additional experimental results obtained with fictional prompts in the camera-ready version. We believed that using real-entity prompts would be a suitable setting as a first step, considering the practical usage of LLMs (i.e., people don’t query LLMs with fictional prompt), but again your suggestion is insightful for scientific analysis.
>
> We recognize that the descriptions of our controlled task in Table 1 are incorrect. Specifically, under the FIC, FICNEG, FICNEG-O settings, the variables should be those representing real ones (e.g., a, b, etc.) instead of the current ones (e.g., $\alpha$, $\beta$, etc.); that is, your understanding and the examples shown in Tables 14–18 are correct (few-shot examples have real entities). We’ll correct the table and apologize for the potential confusion.
>
> **Mismatched caption in the appendix (Tables 5 and 6)**
>
> We will ensure to make the corrections in the camera-ready version. We’re sorry for the mismatched caption of Tables 5 and 6 in Appendix E.

---

### Meta-Review · Area_Chair_S7eG · 2023-09-18

**Recommendation:** 5

**Metareview:**

The reviewers agreed that this is a solid paper with well-structured and detailed experiments, as well as clear articulation of the findings. They also found that the results around different LLMs being differenly successful in dealing with negation to be interesting. The work is timely as it helps us better understand today's popular LMs with impressive (but error-prone) chain-of-thought abilities.

The reviewers pointed out some ways to improve presentation (e.g., splitting Tables 2 and 3), which the authors acknowledged and agree to.

---

### Decision · Program_Chairs · 2023-10-07

**Decision:**

Accept-Main

**Comment:**

The reviewers agreed that this is a solid paper with well-structured and detailed experiments, as well as clear articulation of the findings. They also found that the results around different LLMs being differenly successful in dealing with negation to be interesting. The work is timely as it helps us better understand today's popular LMs with impressive (but error-prone) chain-of-thought abilities.

The reviewers pointed out some ways to improve presentation (e.g., splitting Tables 2 and 3), which the authors acknowledged and agree to.